https://doi.org/10.1038/s41467-021-21477-w · **OPEN**

# Dysregulation of REV-ERBα impairs GABAergic function and promotes epileptic seizures in preclinical models

Tianpeng Zhang[1,2,3], Fangjun Yu[2], Haiman Xu[2], Min Chen[2], Xun Chen[2], Lianxia Guo[2], Cui Zhou[2], Yuting Xu[4], Fei Wang[2], Jiandong Yu [5✉] & Baojian Wu [1,2✉]

To design potentially more effective therapies, we need to further understand the mechanisms underlying epilepsy. Here, we uncover the role of Rev-erbα in circadian regulation of epileptic seizures. We first show up-regulation of REV-ERBα/Rev-erbα in brain tissues from patients with epilepsy and a mouse model. Ablation or pharmacological modulation of Rev-erbα in mice decreases the susceptibility to acute and chronic seizures, and abolishes diurnal rhythmicity in seizure severity, whereas activation of Rev-erbα increases the animal susceptibility. Rev-erbα ablation or antagonism also leads to prolonged spontaneous inhibitory postsynaptic currents and elevated frequency in the mouse hippocampus, indicating enhanced GABAergic signaling. We also identify the transporters Slc6a1 and Slc6a11 as regulators of Rev-erbα-mediated clearance of GABA. Mechanistically, Rev-erbα promotes the expressions of Slc6a1 and Slc6a11 through transcriptional repression of E4bp4. Our findings propose Rev-erbα as a regulator of synaptic function at the crosstalk between pathways regulating the circadian clock and epilepsy.

[1] Institute of Molecular Rhythm and Metabolism, Guangzhou University of Chinese Medicine, Guangzhou, China. [2] College of Pharmacy, Jinan University, Guangzhou, China. [3] Integrated Chinese and Western Medicine Postdoctoral research station, Jinan University, Guangzhou, China. [4] Department of Neurosurgery, Zhujiang Hospital, Southern Medical University, Guangzhou, China. [5] Guangdong-Hongkong-Macau Institute of CNS Regeneration, Jinan University, Guangzhou, China. ✉email: yujiandong@jnu.edu.cn; bj.wu@hotmail.com

Temporal lobe epilepsy (TLE) is the most common form of epilepsy, a serious chronic neurological disorder characterized by recurrent unprovoked seizures that affects ~50 million people worldwide[1]. In about one-third of patients, epilepsy is refractory to antiepileptic drugs (AEDs), and surgical removal of the epileptic foci is suitable only for a portion of them[2]. Although AEDs can reduce or eliminate seizures for the more fortunate patients, these medicines are associated with diverse and troublesome side effects, including weight gain, metabolic acidosis, hepatotoxicity, and movement disorders[3]. Therefore, understanding the molecular events underlying the occurrence of seizures (and epileptogenesis) is necessary and more effective therapeutic agents are desperately needed.

γ-Aminobutyric acid (GABA), a principal inhibitory neurotransmitter in the brain, functions to maintain inhibitory tone that counterbalances neuronal excitation (seizures may ensue when the balance is perturbed)[4]. GABA is formed within GABAergic axon terminals by two sequential reactions catalyzed by GABA α-oxoglutarate transaminase and glutamic acid decarboxylase. It is released into the synaptic cleft and acts on GABA receptors[5]. GABA is rapidly removed by reuptake into presynaptic neurons and glia mediated by Slc6a1 (Gat1) and Slc6a11 (Gat3), respectively[6]. Mounting evidence supports a critical role of GABA in epilepsy and in epileptogenesis[4,7]. AEDs such as benzodiazepines and barbiturates act to promote the binding of GABA to GABA receptors, therefore, enhancing GABAergic function[8]. Also, tiagabine and vigabatrin are two AEDs that increase synaptic GABA level to inhibit the seizure activity[9].

Circadian rhythmicity in epileptic seizures has long been recognized in humans and in rodents[10–12]. Seizures in rodent models have a tendency to occur more frequently in the light phase than in the dark phase[10,11,13,14]. These diurnal events suggest a potential role of circadian clock in regulation of seizure occurrence. Circadian clock is an autoregulatory feedback loop system consisting of transcriptional activators (e.g., Bmal1 and Clock) and repressors (e.g., Cry, Per, and Rev-erbα)[15]. Single deletion of core clock genes (Bmal1 and Clock) in mice has been associated with an increased susceptibility to epilepsy[16,17]. Altered expression of clock genes is observed in human epileptic tissues[16]. All these support involvement of circadian clock in the development of epilepsy. However, the molecular mechanisms for circadian clock-controlled epilepsy remain poorly explored, and how seizure rhythmicity is generated is unanswered.

Here, we investigated a potential role of Rev-erbα, a ligand-responsive clock component, in circadian regulation of epileptic seizures. We first found that REV-ERBα/Rev-erbα was upregulated in human and mouse epileptic tissues. Rev-erbα ablation or antagonism reduced the sensitivity of mice to acute and chronic seizures, and abolished the diurnal rhythmicity in seizure severity, whereas Rev-erbα activation increased the animal sensitivity. Mechanistically, Rev-erbα reduced GABA-mediated inhibition of neurons by promoting Slc6a1/Slc6a11 expressions and reuptake of GABA. Promotion of Slc6a1/Slc6a11 expressions was attained through transcriptional repression of E4bp4.

## Results

### Rev-erbα is dysregulated in human and mouse epileptic tissues.
We collected surgical samples (hippocampus and temporal cortex) from patients with TLE and with glioma (as a control). TLE patients were diagnosed by using high-resolution magnetic resonance imaging, positron emission tomography/computed tomography, and electroencephalography (EEG; Fig. S1). TLE patients showed a higher expression of REV-ERBα (a circadian clock component) in both hippocampus and temporal cortex, as compared to control patients (Fig. 1a, b). By contrast, other circadian clock genes, such as BMAL1, CLOCK, E4BP4, and DBP (direct or indirect target genes of REV-ERBα) were downregulated in TLE patients (Fig. S2). Immunofluorescence experiments confirmed the presence of REV-ERBα in neurons and glial cells, and increased REV-ERBα expression in TLE patients (Fig.1c–e and Fig. S3A). Rev-erbα mRNA and protein were found in main brain sections (including hippocampus, cortex, cerebellum, nigra, striatum, and hypothalamus) in normal mice (Fig. S4A, B). Rev-erbα was localized in neurons and glial cells in mouse hippocampus (Fig. S5). In line with the human data, we observed increased Rev-erbα expression in both hippocampus and cortex in kainic acid (KA)-treated mice (a model of status epilepticus) that was accompanied by reductions in Bmal1, Clock, E4bp4, and Dbp mRNAs (Fig. 1f, g). These data suggested potential involvement of REV-ERBα in epileptic seizures.

To identify the circadian pattern of epileptic seizures, mice were treated with KA at each of six different circadian time points (i.e., ZT2, ZT6, ZT10, ZT14, ZT18, and ZT22). Seizure occurrence (or disease severity) as reflected by the epileptic activity and mortality rate displayed a robust diurnal rhythm with a zenith at around ZT6 and a nadir at around ZT18 (Fig. 1h, i). This diurnal pattern paralleled with circadian protein expression of Rev-erbα in mouse hippocampus and cortex, supporting a connection between Rev-erbα and epileptic seizures (Fig. 1h, i and Fig. S4C). Taken together, Rev-erbα may act as a circadian regulator of epileptic seizures.

### Loss of Rev-erbα reduces acute seizures in mice.
We next investigated the role of Rev-erbα in epileptic seizures using gene knockout (Rev-erbα$^{-/-}$ or KO) mice. Rev-erbα ablation retarded the progression of behavioral seizure stages and reduced the seizure severity in the KA model (Fig. 2a, b). Onset of first seizure was prolonged and seizure duration was decreased in Rev-erbα$^{-/-}$ mice (Fig. 2b). Also, Rev-erbα ablation downregulated the frequency of KA-induced seizures based on EEG recordings (Fig. 2c). Progressive neuron loss and gliosis are common pathological hallmarks of TLE[1]. KA-challenged Rev-erbα$^{-/-}$ mice showed a lower level of neuron death in the hippocampus as compared to wild-type mice, according to Fluoro-Jade-B (FJB) and TUNEL staining (Fig. 2d, e and Fig. S3B). This was accompanied by a higher number of living neurons and reduced astrogliosis in Rev-erbα$^{-/-}$ mice (Fig. 2f and Fig. S3B). Moreover, the effects of Rev-erbα ablation on KA-induced seizures were circadian time-dependent. Rev-erbα ablation markedly attenuated the seizures when KA was injected at ZT6, while having no significant effects when the chemical was injected at ZT18 (Fig. 2g–i). As a consequence, the time difference in disease severity was lost in Rev-erbα$^{-/-}$ mice (Fig. 2g–i). In addition, we examined the effects of Rev-erbα on KA-induced acute seizures using a chronic jet lag model (mice were subjected to 8-h light advance every 2 days for 8 weeks; Fig. S6A). As expected, jet-lagged mice showed disrupted diurnal expressions of clock genes in the hippocampus and cortex, particularly, Rev-erbα was markedly downregulated at ZT6 but unaltered at ZT18 (Fig. S6B). In line with the Rev-erbα expression changes, jet-lagged mice were more resistant to KA-induced acute seizures at ZT6 (Fig. S6C, D). Also, time dependency of seizure severity ceased to exist in jet-lagged mice (Fig. S6C, D). Taken together, these data defined a critical role of Rev-erbα in the circadian regulation of acute seizures. It was noteworthy that genetic deletion of Rev-erbβ (a Rev-erbα paralogue) might have no effects on KA-induced acute seizures (only one KA dose was tested), probably precluding a role of this receptor in regulating seizures (Fig. S7).

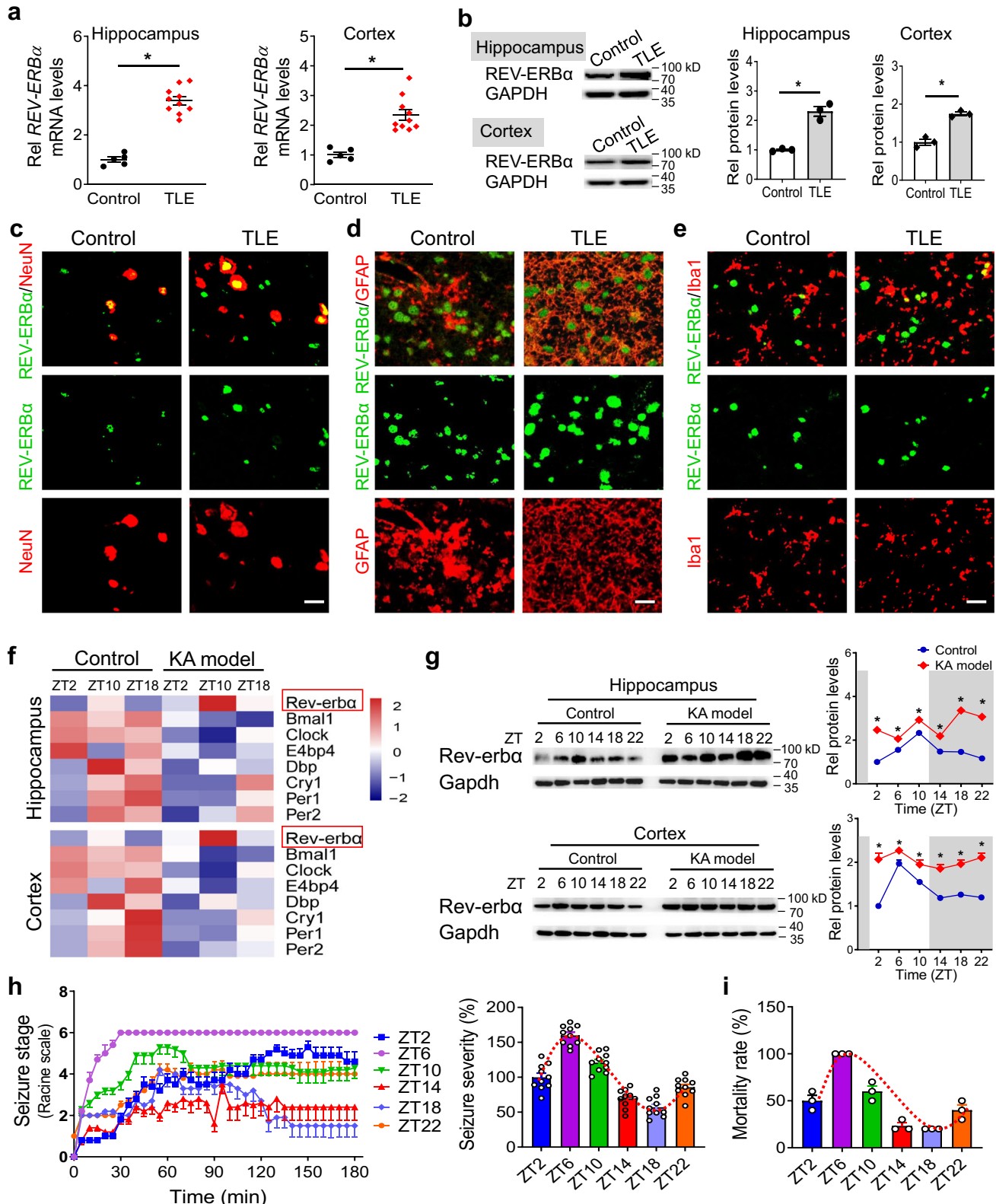

**Small-molecule targeting of Rev-erbα alleviates acute and chronic seizures.** Due to a potential role of Rev-erbα in epileptic seizures, we explored whether it can be targeted by small molecules to treat the disease. We confirmed a high drug distribution to the brain after dosing of SR8278 (a specific antagonist of Rev-erbα) to mice (Fig. S8). SR8278 treatment delayed the progression of behavioral seizure stages and reduced the seizure severity in KA-treated mice (wild-type; Fig. 3a, b). Accordingly, onset of first seizure was prolonged and seizure duration was reduced (Fig. 3b). Also, seizure frequency was downregulated by SR8278 according to EEG recordings (Fig. 3c). Furthermore, FJB and TUNEL staining revealed a lower rate of neuron death in SR8278-treated mice (Fig. 3d, e and Fig. S3C) that coincided with a higher number of living neurons and reduced astrogliosis (Fig. 3f and

**Fig. 1 Rev-erbα is dysregulated in human and mouse epileptic tissues. a** mRNA expression of *REV-ERBα* in hippocampus and cortex from TLE patients ($n = 10$ biologically independent samples) and controls ($n = 5$ biologically independent samples). Two-sided $t$ test $p$ values: $p < 0.0001$ (hippocampus); $p = 0.003$ (cortex). **b** Protein expression of REV-ERBα in hippocampus and temporal cortex from TLE patients and controls. TLE and control samples were pooled for western blotting analysis. Each western blot is representative of three independent experiments. Western blot strips (a target protein and a loading control) were cut from one gel. Two-sided $t$ test $p$ values: $p = 0.0013$ (hippocampus); $p = 0.0013$ (cortex). **c–e** REV-ERBα colocalizes with neuronal marker NeuN, astrocytic marker GFAP, and microglial marker Iba1 in human epileptic and control tissues. Similar results were obtained in three independent experiments. Scale bar = 20 μm. **f** Heatmap of clock gene transcripts in hippocampus and cortex from kainic acid (KA)-treated and control mice. **g** Rev-erbα protein levels in hippocampus and cortex from KA-treated mice ($n = 6$ biologically independent samples) and controls ($n = 6$ biologically independent samples). Western blot strips (a target protein and a loading control) were cut from one gel. P values (hippocampus, from left to right): $p = <0.0001, 0.0015, 0.0050, 0.0002, <0.0001$, and $0.0001$ (two-way ANOVA and Bonferroni post hoc test). P values (cortex, from left to right): $p = 0.0017, 0.0405, 0.0247, 0.0023, 0.0027$, and $0.0016$ (two-way ANOVA and Bonferroni post hoc test). **h** Seizure stages of wild-type (WT) mice ($n = 10$ per group) treated with KA (20 mg/kg, i.p.) at each of six circadian time points (ZT2, ZT6, ZT10, ZT14, ZT18, and ZT22). **i** Mortality rates of WT mice treated with KA (20 mg/kg, i.p.) at each of six circadian time points (ZT2, ZT6, ZT10, ZT14, ZT18, and ZT22). Three independent experiments (each with ten mice) were performed to determine the mortality rate. In **a**, **b** and **g–i**, data are presented as mean ± SEM. * represents a $p$ value of <0.05. TLE temporal lobe epilepsy, Rel relative, ZT zeitgeber time.

Fig. S3C). However, these protective effects of SR8278 on acute seizures were lost in *Rev-erbα*$^{-/-}$ mice (Fig. 3g, h), confirming a Rev-erbα-dependent drug action. Supporting this, mice treated with SR9009 (a known agonist of Rev-erbα) were more sensitive to KA-induced acute seizures, as evidenced by more rapid seizure onset, prolonged seizure duration, and exacerbated seizure severity (Fig. S9). In addition, we assessed the Rev-erbα effects using the pilocarpine model of chronic spontaneous seizures (Fig. S10A). Rev-erbα ablation or antagonism (by SR8278) led to a decreased number of seizures and reduced seizure severity, as well as alleviated neuronal damage (Figs. S10 and S11). By contrast, Rev-erbα activation (by SR9009) exacerbated pilocarpine-induced chronic seizures in mice (Fig. S11). All these data indicate that small-molecule antagonism of Rev-erbα is therapeutically beneficial for the management of epileptic seizures.

**Rev-erbα ablation alleviates TLE in a hippocampal kindling model.** *Rev-erbα*$^{-/-}$ and wild-type mice were subjected to hippocampal kindling to induce TLE (Fig. 4a). TLE was less severe in *Rev-erbα*$^{-/-}$ mice than in wild-type mice (Fig. 4b–d). *Rev-erbα*$^{-/-}$ mice showed delayed progression of behavioral seizure stages and shortened afterdischarge duration (ADD), as compared to wild-type mice (Fig. 4b). In fact, *Rev-erbα*$^{-/-}$ mice were resistant to the TLE development as evidenced by an increased number of stimulations required to reach generalized seizure (GS, stage ≥4; Fig. 4c). Supporting this, seizure frequency was lower in *Rev-erbα*$^{-/-}$ mice than in wild-type mice upon hippocampal kindling, according to EEG recordings and power spectra (Fig. 4d and Fig. S12). In addition, we assessed the effects of Rev-erbα antagonism (by SR8278) on kindling-induced TLE in wild-type mice (Fig. 4e). SR8278 was able to reduce the seizure severity and to decrease the ADD level (Fig. 4f). The number of stimulations required to reach GS was increased for SR8278-treated mice (Fig. 4g). In the meantime, SR8278 decreased the epileptic activity in mice (Fig. 4h and Fig. S12). Taken together, targeted inhibition of Rev-erbα activity can ameliorate the epileptic seizures in the hippocampal kindling model.

**Rev-erbα regulates spontaneous inhibitory postsynaptic currents and GABA reuptake.** To determine the functional relevance of Rev-erbα at a cellular level, we performed whole-cell recordings in the dentate gyrus granule cells (DGGC) of hippocampus slices derived from *Rev-erbα*$^{-/-}$ and wild-type mice. The intrinsic properties of DGGC were not different between *Rev-erbα*$^{-/-}$ and wild-type mice (Supplementary Table 1). The frequency and decay time of spontaneous inhibitory postsynaptic currents (sIPSCs) were increased, but the amplitude and rise time

were unaltered in *Rev-erbα*$^{-/-}$ mice as compared to wild-type mice (Fig. 5a–c). By contrast, Rev-erbα ablation had no effects on miniature excitatory postsynaptic currents (mEPSCs; Fig. S13). We found that sIPSCs had increased frequency and decay time, but unaltered amplitude and rise time in SR8278-treated wild-type mice (Fig. 5d–f). Also, tonic GABA currents were enhanced in *Rev-erbα*$^{-/-}$ mice as compared to wild-type mice (Fig. S13). These data indicated potential regulation of synaptic transmission (GABAergic signaling) by Rev-erbα. Furthermore, we compared hippocampus and cortex transcriptomes between *Rev-erbα*$^{-/-}$ and wild-type mice. As expected, gene alterations in *Rev-erbα*$^{-/-}$ mice were more extensive at ZT6 than at ZT18 (Fig. 5g, h). Of note, *Slc6a1* and *Slc6a11* are two genes most differentially expressed (Fig. 5h). Proteins coded by these two genes mediate reuptake processes of GABA to presynaptic neurons and glia, respectively (Fig. 5i). This suggested involvement of Rev-erbα in the regulation of GABA reuptake (clearance) that is consistent with a prolonged decay time in sIPSCs in Rev-erbα-deficient neurons. Supporting this, Rev-erbα ablation significantly reduced the cellular uptake of GABA-d6 in three independent sets of transport experiments with cerebral cortex slices, primary hippocampus, and cortex neurons (Fig. 5j, k). Further, the deregulation of sIPSCs and tonic GABA currents in *Rev-erbα*$^{-/-}$ mice disappeared when chemical inhibitors (NO-711 and SNAP-5114) of Slc6a1 and Slc6a11 were applied (Figs. S13 and 14). It was noted that the plasma, hippocampus, and cortex concentrations of KA after systemic administration were unchanged in *Rev-erbα*$^{-/-}$ mice, and that Rev-erbα ablation had no effects on KA receptor expression (Fig. S15). Altogether, Rev-erbα regulates GABAergic signaling probably through promoting Slc6a1 and Slc6a11 activities.

**Rev-erbα positively regulates Slc6a1 and Slc6a11 expressions via repression of E4bp4.** We analyzed the hippocampus and cortex expressions of GABA-related genes in *Rev-erbα*$^{-/-}$ mice versus in wild-type mice. Rev-erbα ablation downregulated expression levels of Slc6a1 and Slc6a11 in mice, whereas all other GABA-related genes were unaffected (Fig. 6a–d and Fig. S13C). Of note, diurnal rhythms of Slc6a1 and Slc6a11 expressions were blunted in *Rev-erbα*$^{-/-}$ mice (Fig. 6a–d). Consistent with a positive regulatory effect, overexpression of Rev-erbα increased *Slc6a1* and *Slc6a11* expressions in Neuro-2a cells, primary hippocampal neurons, and primary cortex neurons, whereas knockdown of Rev-erbα reduced their cellular expressions (Fig. 6e–g). Also, the Rev-erbα antagonist SR8278 dose-dependently reduced *Slc6a1 and Slc6a11* mRNAs in primary hippocampus and cortex neurons (Fig. 6h). We also examined glutamate (a major excitatory neurotransmitter)-related genes,

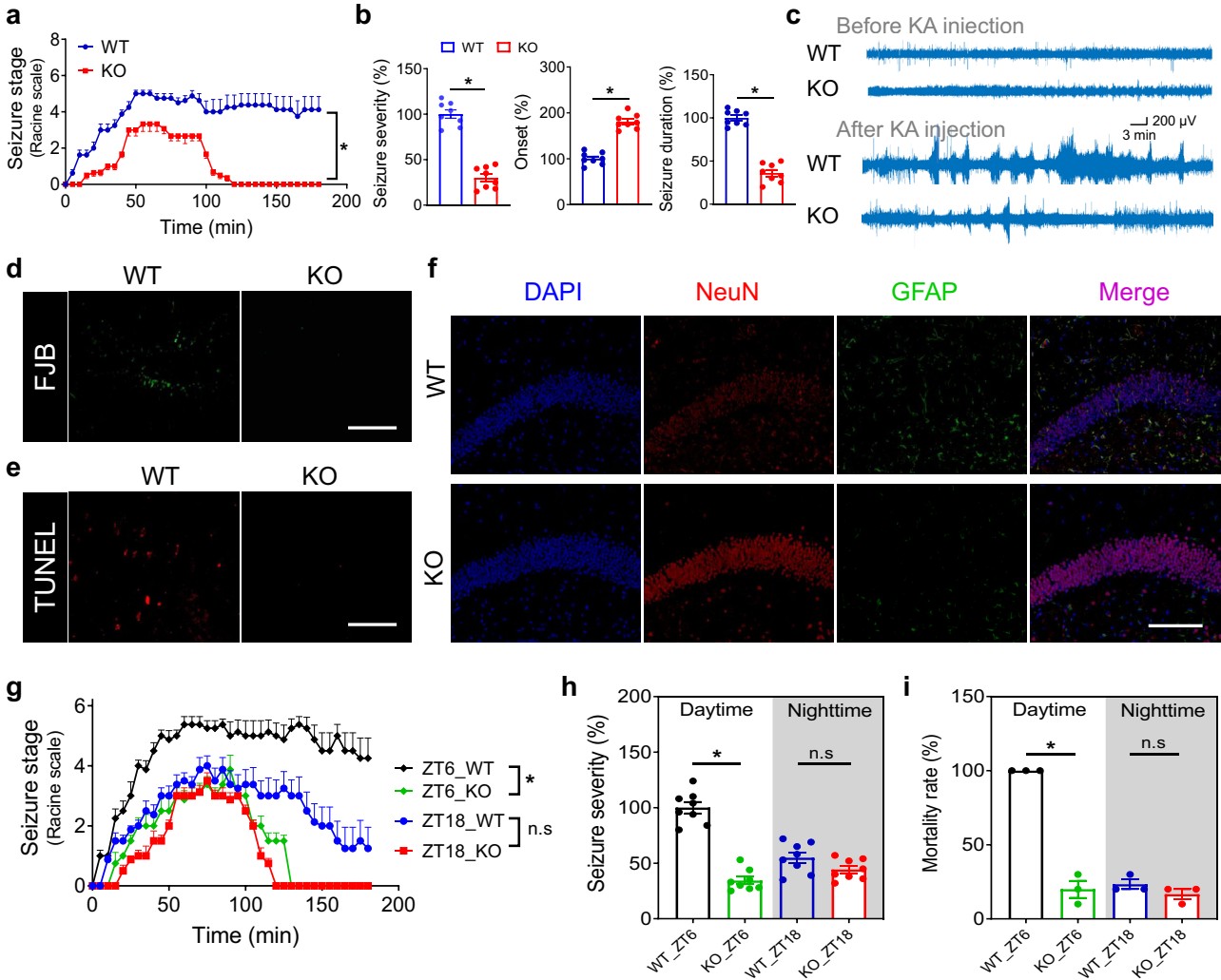

**Fig. 2 Loss of Rev-erbα reduces acute seizures in mice. a** Seizure stages of *Rev-erbα*$^{-/-}$ (KO) mice ($n = 8$ biologically independent samples) and wild-type (WT) mice ($n = 8$ biologically independent samples) injected with kainic acid (KA, 20 mg/kg) at ZT6. $P < 0.0001$ (two-sided Kruskal–Wallis test). **b** Seizure parameters (severity, onset, and duration) of *Rev-erbα*$^{-/-}$ mice ($n = 8$ biologically independent samples) and WT mice ($n = 8$ biologically independent samples) injected with KA (20 mg/kg) at ZT6. Two-sided $t$ test $p$ values: $p < 0.0001$ (seizure severity); $p < 0.0001$ (onset); $p < 0.0001$ (seizure duration). **c** Representative EEG tracings in mice before (3 h) and after (3 h) KA injection (20 mg/kg). **d** FJB staining of hippocampus sampled at 24 h after injection of KA (20 mg/kg) to *Rev-erbα*$^{-/-}$ and WT mice. **e** TUNEL staining of hippocampus sampled at 24 h after injection of KA (20 mg/kg). **f** NeuN and GFAP staining of hippocampus at 24 h after KA injection (20 mg/kg). **g** Seizure stages of *Rev-erbα*$^{-/-}$ mice ($n = 8$ biologically independent samples) and WT mice ($n = 8$ biologically independent samples) injected with KA (20 mg/kg) at ZT6 and ZT18. Two-sided Kruskal–Wallis test $p$ values: $p < 0.0001$ (ZT6); $p = 0.1213$ (ZT18). **h** Seizure severity of *Rev-erbα*$^{-/-}$ mice ($n = 8$ biologically independent samples) and WT mice ($n = 8$ biologically independent samples) injected with KA (20 mg/kg) at ZT6 and ZT18. $P < 0.0001$ (ZT6); $p = 0.0893$ (ZT18; two-way ANOVA and Bonferroni post hoc test). **i** Mortality rates of *Rev-erbα*$^{-/-}$ and WT mice injected with KA (20 mg/kg) at ZT6 and ZT18. Three independent experiments (each with ten mice) for each group were performed to determine the mortality rate. $P = 0.0002$ (ZT6); $p = 0.2302$ (ZT18; two-way ANOVA and Bonferroni post hoc test). Scale bar = 50 μm. In **a**, **b** and **g–i**, data are presented as mean ± SEM. For **d–f**, similar results were obtained in three independent experiments. * represents a $p$ value of <0.05. FJB Fluoro-Jade-B, n.s no significant, ZT zeitgeber time.

and found none of these gene were regulated by Rev-erbα (Figs. S16 and 17). Thereby, the role of glutamate signaling in Rev-erbα regulation of epilepsy was excluded.

Contrasting with functioning as a transcription repressor[18], Rev-erbα positively regulated the transporters Slc6a1 and Slc6a11. It was hypothesized that an indirect mechanism involving a negative regulator was necessary for Rev-erbα regulation of Slc6a1/Slc6a11. E4bp4, a known transcriptional target of Rev-erbα, has been shown to be a repressor of multiple transporters, such as P-gp (P-glycoprotein) and Mrp2 (multidrug resistance-associated protein 2)[19,20]. We thus investigated a potential mediating role of E4bp4 in connecting Rev-erbα to

Slc6a1/Slc6a11. As expected, Rev-erbα ablation resulted in the increased protein expression of E4bp4 in mouse hippocampus and cortex (Fig. 6c, d). In fact, Rev-erbα trans-represses *E4bp4* through direct binding to a specific response element (called RevRE, −2850/−2824 bp) in the gene's promoter[21].

E4bp4 ablation significantly increased the hippocampus and cortex expressions of both Slc6a1 and Slc6a11 in mice, suggesting E4bp4 as a repressor of the two transporters (Fig. 7a, b and Fig. S18). Supporting this, E4bp4 silencing increased the expressions of *Slc6a1* and *Slc6a11* in Neuro-2a cells, primary hippocampal neurons, and primary cortex neurons, whereas the overexpression of E4bp4 reduced cellular expressions of these

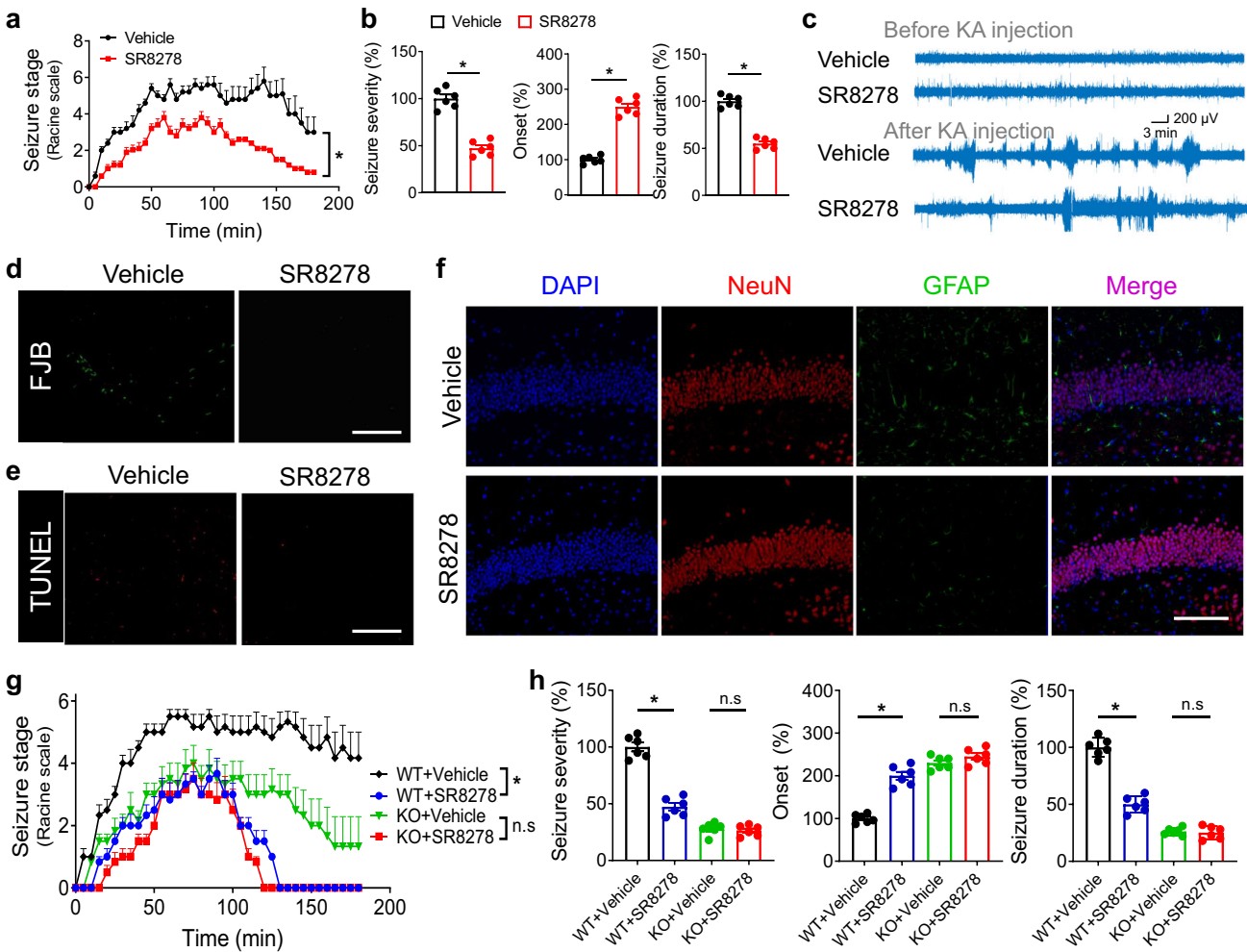

**Fig. 3 Small-molecule targeting of Rev-erbα alleviates acute and chronic seizures. a** Effects of SR8278 pretreatment (25 mg/kg, i.p.) on seizure stages of wild-type (WT) mice induced by kainic acid (KA, 20 mg/kg, i.p.) at ZT6. $P < 0.0001$ (two-sided Kruskal–Wallis test). **b** Effects of SR8278 pretreatment on seizure parameters (severity, onset, and duration) of KA-induced acute seizure mice. Two-sided $t$ test $p$ values: $p < 0.0001$ (seizure severity); $p < 0.0001$ (onset); $p < 0.0001$ (seizure duration). **c** Representative EEG tracings in mice (pretreated with SR8278 or vehicle) before (3 h) and after (3 h) KA injection (20 mg/kg, i.p.). **d** FJB staining of hippocampus at 24 h after KA (20 mg/kg, i.p.) administration to WT mice pretreated with SR8278 (25 mg/kg, i.p.) or vehicle. Similar results were obtained in three independent experiments. **e** TUNEL staining of hippocampus at 24 h after KA (20 mg/kg, i.p.) administration to WT mice pretreated with SR8278 (25 mg/kg, i.p.) or vehicle. Similar results were obtained in three independent experiments. **f** Hippocampus NeuN and GFAP staining in mice (pretreated with SR8278 or vehicle) at 24 h after treatment with KA (20 mg/kg, i.p.). Similar results were obtained in three independent experiments. **g** Effects of SR8278 pretreatment (25 mg/kg, i.p.) on seizure stages of Rev-erbα$^{-/-}$ and WT mice induced by KA (20 mg/kg, i.p.) at ZT6. Two-sided Kruskal–Wallis test $p$ values: $p < 0.0001$ (ZT6); $p = 0.1027$ (ZT18). **h** Effects of SR8278 pretreatment (25 mg/kg, i.p.) on seizure severity of Rev-erbα$^{-/-}$ and WT mice induced by KA (20 mg/kg, i.p.) at ZT6. $P < 0.0001$ (WT, seizure severity); $p = 0.0893$ (KO, seizure severity); $p < 0.0001$ (WT, onset); $p = 0.0635$ (KO, onset); $p < 0.0001$ (WT, seizure duration); $p = 0.0643$ (KO, seizure duration; two-way ANOVA followed by Bonferroni post hoc test). Scale bar = 50 μm. In **a**, **b**, **g** and **h**, data are presented as mean ± SEM. $n = 6$ mice per group. * represents a $p$ value of <0.05. FJB Fluoro-Jade-B, n.s. no significant.

transporters (Fig. 7c–e). Since E4bp4 functions as a transcriptional factor, we tested whether it regulated Slc6a1 and Slc6a11 via a transcriptional mechanism. E4bp4 dose-dependently inhibited the promoter activities of Slc6a1 (−2000/+38 bp) and Slc6a11 (−2500/+36 bp; Fig. 7f). Promoter analysis predicted a potential D-box (a motif for E4bp4 binding, −1544/−1535 bp for Slc6a1 and −1547/−1535 bp for Slc6a11) in the promoter regions of Slc6a1 and Slc6a11 (Fig. 7f). Mutation experiments revealed that the predicted D-box elements were essential for E4bp4 actions (Fig. 7f). ChIP assays confirmed direct interactions of E4bp4 protein with Slc6a1-D-box and Slc6a11-D-box (Fig. 7g). Therefore, E4bp4 trans-repressed Slc6a1 and Slc6a11 via direct binding to a D-box element in their promoters. In addition, Rev-erbα increased the promoter activities of Slc6a1 and Slc6a11 in luciferase reporter assays (Fig. 7h). However, knockdown of E4bp4 attenuated the activation effects of Rev-erbα on Slc6a1 and Slc6a11 transcription, indicating an E4bp4-dependent regulation mechanism (Fig. 7h). Supporting this, E4bp4 ablation sensitized mice to KA-induced acute seizure (i.e., exacerbated seizure severity in E4bp4$^{-/-}$ mice as compared to wild-type mice; Fig. 7i, j). Taken together, Rev-erbα positively regulates Slc6a1 and Slc6a11 expression via transcriptional repression of E4bp4.

## Discussion

We have defined a functional role of the clock component Rev-erbα in epileptic seizures, highlighting the crosstalk between circadian biology and epileptic seizures. Combined with prior reports,

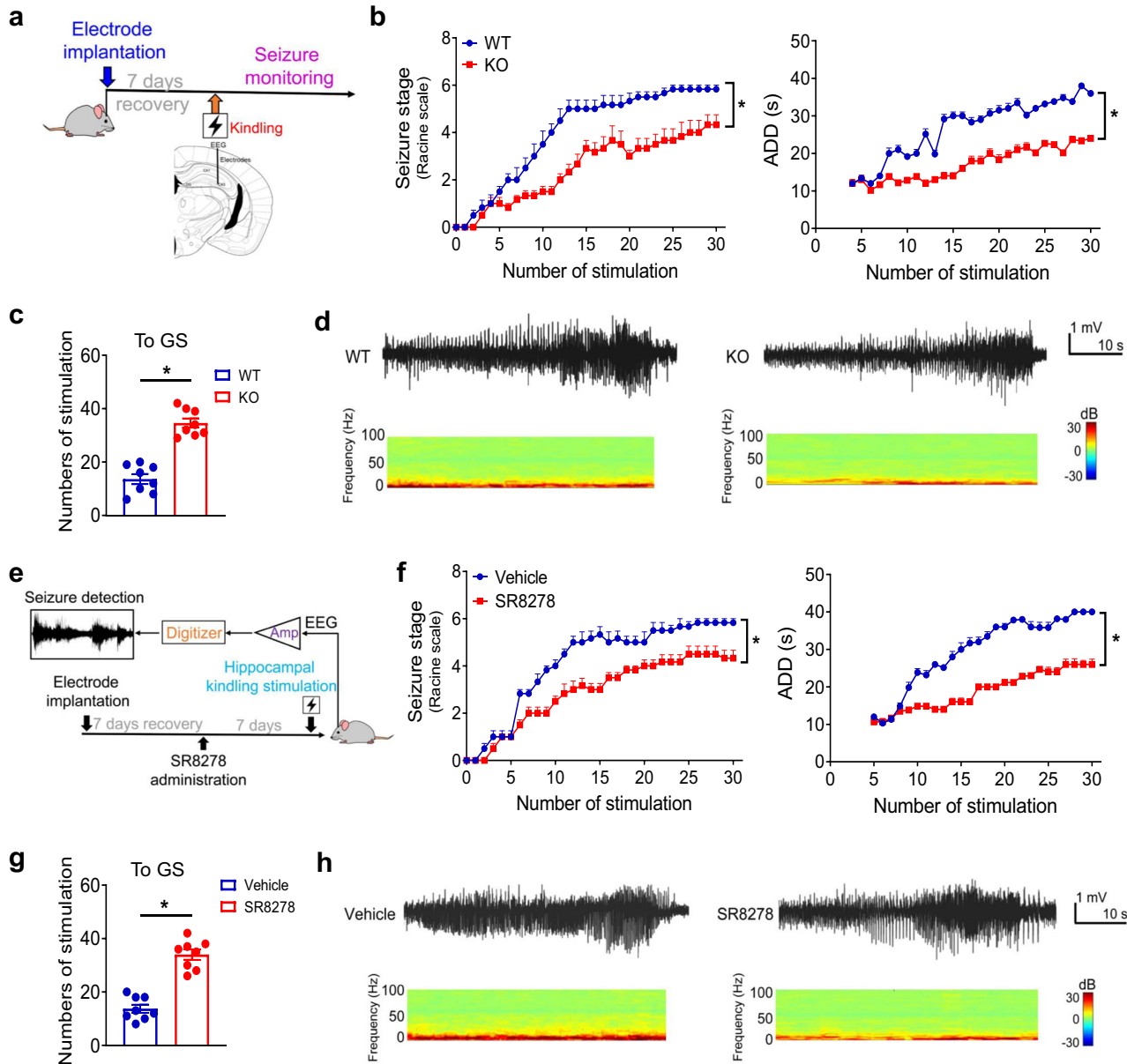

**Fig. 4 Rev-erbα ablation alleviates TLE in a hippocampal kindling model. a** Experimental scheme for establishment of kindling-induced TLE model with *Rev-erbα*−/− (KO) and wild-type (WT) mice. The image was created by T. Zhang. **b** Seizure stages and ADD (afterdischarge duration) of *Rev-erbα*−/− and WT mice after repeated kindling stimulations. *P* values: <0.0001 (seizure stages), <0.0001 (ADD; two-sided Kruskal–Wallis test). **c** Numbers of stimulation required to reach generalized seizure of *Rev-erbα*−/− and WT mice. *P* < 0.0001 (two-sided *t* test). **d** EEG recordings and power spectra of *Rev-erbα*−/− and WT mice in a hippocampal kindling model. Red indicates high relative power and blue indicates low power of EEG frequency bands, as shown in the scale bar. **e** Experimental scheme for SR8278 (25 mg/kg, i.p.) pretreatment and kindling stimulations with WT mice. **f** Effects of SR8278 on seizure stages and ADD of WT mice in a hippocampal kindling model. *P* values: <0.0001 (seizure stages); <0.0001 (ADD; two-sided Kruskal–Wallis test). **g** Effects of SR8278 on numbers of stimulation required to reach generalized seizure. *P* < 0.0001 (two-sided *t* test). **h** EEG recordings and power spectra of kindling-induced TLE mice pretreated with SR8278 or vehicle. Red indicates high relative power and blue indicates low power of EEG frequency bands, as shown in the scale bar. In **b**, **c**, **f** and **g**, data are presented as mean ± SEM. *n* = 6 mice per group. * represents a *p* value of <0.05. ADD afterdischarge duration, GS generalized seizure.

circadian clock appears to regulate epileptic seizures via pleiotropic effects, including the inhibition of GABAergic function by Rev-erbα/E4bp4, activation of pyridoxal kinase by TEF (thyrotroph embryonic factor), and alteration of cortical circuits by Clock[16,22]. We observed dysregulation of other clock genes (Bmal1, Clock, and Dbp) in epileptic tissues in addition to Rev-erbα (Fig. 1). Expression change in *Rev-erbα* (upregulation) may precede that in *Bmal1* (downregulation) and in *Clock* (downregulation) because Rev-erbα is a repressor of Bmal1/Clock, whereas Bmal1/Clock are activators

of Rev-erbα[18,23,24]. We thus propose that Rev-erbα is a fundamental factor linking circadian clock to epileptic seizures. However, whether and how epilepsy regulates Rev-erbα remain largely unaddressed although the regulatory mechanisms for circadian clock-controlled epileptic seizures are becoming clearer. It is speculated that epilepsy-driven inflammations may attack circadian clock because inflammatory cytokines (e.g., TNF-α and IL-1β) are modifiers of clock genes, such as *Rev-erbα* and *Per2*[25–27]. The finding that Rev-erbα promotes epileptic seizures helps to explain

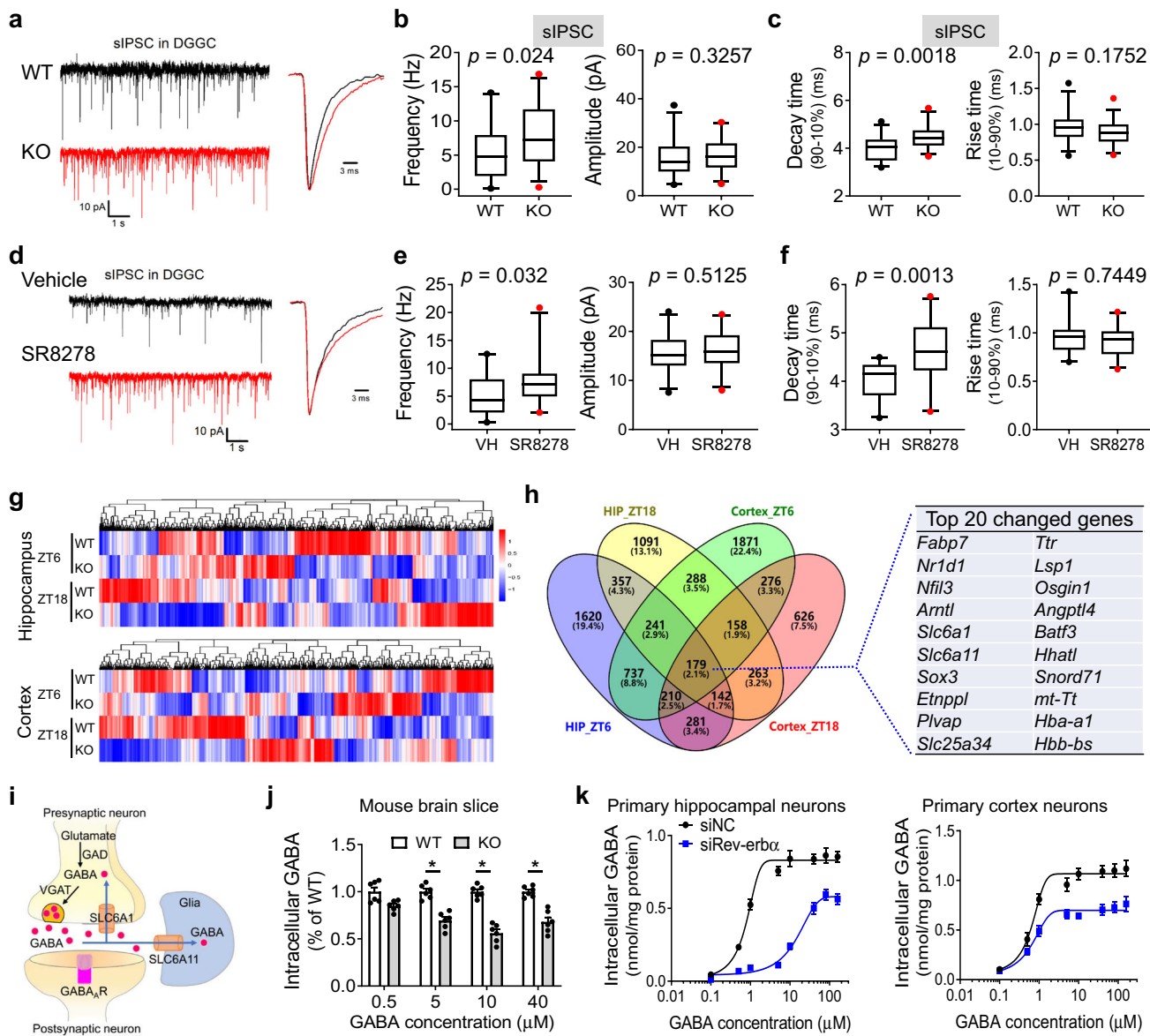

**Fig. 5 Rev-erbα regulates spontaneous inhibitory postsynaptic currents and GABA uptake. a** Representative traces of spontaneous inhibitory postsynaptic currents (sIPSCs) derived from dentate gyrus granule cells of *Rev-erbα*$^{-/-}$ (KO) and wild-type (WT) mice. **b** sIPSC frequency and amplitude for KO (33 cells, six mice) and WT mice (33 cells, six mice). *P* values are shown in the figure (two-sided Mann–Whitney test). **c** sIPSC decay time and rise time for KO (33 cells, six mice) and WT mice (33 cells, six mice). *P* values are shown in the figure (two-sided Mann–Whitney test). **d** Representative traces of sIPSCs derived from the DGGCs of SR8278- or vehicle-treated mice. **e** sIPSC frequency and amplitude for SR8278- or vehicle-treated mice (SR8278 group: 23 cells, five mice; vehicle group: 25 cells, five mice). *P* values are shown in the figure (two-sided Mann–Whitney test). **f** sIPSC decay time and rise time for SR8278 treatment (SR8278 group: 23 cells, five mice; vehicle group: 25 cells, five mice). *P* values are shown in the figure (two-sided Mann–Whitney test). **g** Heatmap of genomic transcripts in hippocampus and cortex of KO and WT mice at ZT6 and ZT18. Red indicates high relative expression and blue indicates low expression of genes, as shown in the scale bar. **h** Venn diagram of the uniquely and commonly changed genes in hippocampus and cortex of *Rev-erbα*$^{-/-}$ and WT mice at ZT6 and ZT18, highlighting the top 20 commonly changed genes. **i** Schematic diagram showing GABA signaling pathway. **j** Cellular uptake of GABA-d6 in brain slices derived from KO and WT mice. *n* = 6 mice per group. Two-sided *t* test *p* values: 0.0019 (5 μM), <0.0001 (10 μM), and 0.0003 (40 μM). **k** Effects of Rev-erbα knockdown on GABA-d6 uptake in primary hippocampus and cortex neurons (*n* = 5 biologically independent samples). In **b**, **c**, **e**, and **f**, data are shown as box-and-whisker with median (middle line), 25th–75th percentiles (box), and 5th and 95th percentile (whiskers), as well as outliers (single points). In **j** and **k**, data are mean ± SEM. * Represents a *p* value of <0.05. DGGC dentate gyrus granule cell, VH vehicle, HIP hippocampus.

some interesting observations in the literature. For instance, melatonin is reported to attenuate seizures in humans and mice[28,29]. This is most likely because the compound is able to downregulate *Rev-erbα* expression[30]. Wulff et al. observed an increased number of hippocampal neurons in *Rev-erbα*$^{-/-}$ mice[31]. This is probably due to decreased seizures and reduced neuronal death caused by Rev-erbα ablation.

We have revealed Rev-erbα as a potential circadian regulator of epileptic activity, supporting a higher frequency of epileptic seizures in humans and in rodents during the light phase, when Rev-erbα is more abundantly expressed in the brain (Fig. S4)[10,12,14,32]. Please note it is assumed that diurnal Rev-erbα profiles in humans are similar to those in baboons[32]. Although Clock and Tef regulate epileptic seizures, they may not contribute or contribute little to

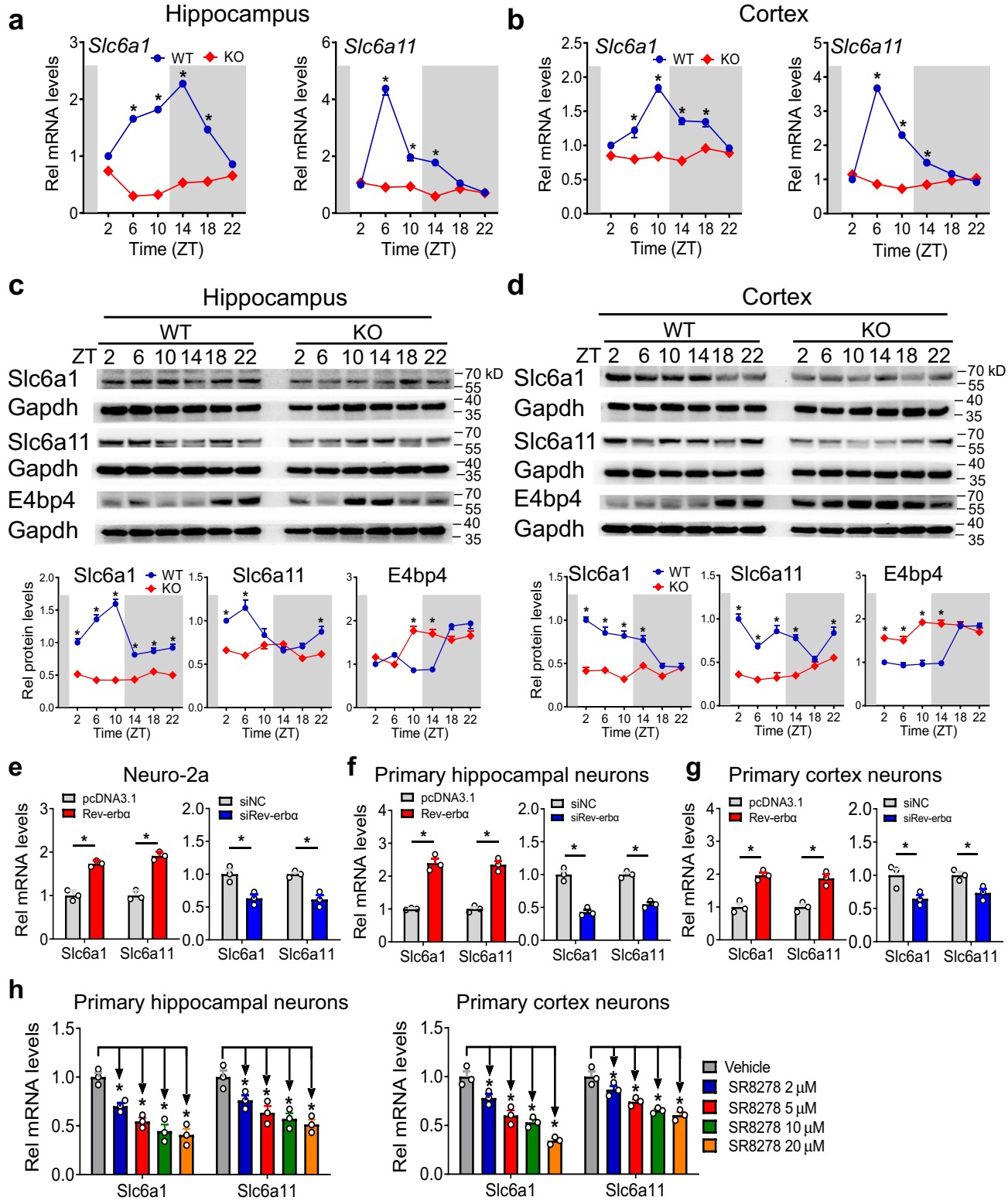

seizure rhythm because Clock protein is circadian time-independent in the cortex and Tef fluctuates mildly[22,33]. Diurnal rhythmicity in epileptic seizures requires chronotherapeutic practices that deliver drugs according to biological rhythms for improved efficacy and reduced toxicity. In fact, dosing time-dependent drug effects have been noted for anticonvulsants, such as carbamazepine[34]. We show that chronopharmacological targeting of Rev-erbα by the antagonist SR8278 alleviates epileptic

seizures. Best dosing time of ZT6-10 corresponds to the highest expression of Rev-erbα (as the drug target) and the highest level of seizure severity. It is noteworthy that SR8278 is not a "pure" antagonist of Rev-erbα, it also acts on Rev-erbβ (a Rev-erbα paralog)[35]. However, it is argued that the antiepileptic effect of SR8278 might be attributed to antagonism of Rev-erbα because the role of Rev-erbβ in epileptic seizures may be limited (Fig. S7). This agrees with the findings that Rev-erbα rather than Rev-erbβ plays a

**Fig. 6 Rev-erbα positively regulates Slc6a1 and Slc6a11 expressions. a** mRNA expressions of *Slc6a1* and *Slc6a11* in the hippocampus of *Rev-erbα*$^{-/-}$ (KO) and wild-type (WT) mice. $n = 6$ mice per group. Data are presented as the fold change in gene expression normalized to *Cyclophilin b* and relative to ZT2 of WT mice. **b** mRNA expressions of *Slc6a1* and *Slc6a11* in the cortex of *Rev-erbα*$^{-/-}$ and WT mice. $n = 6$ mice per group. **c** Protein levels of Slc6a1, Slc6a11, and E4bp4 in the hippocampus of *Rev-erbα*$^{-/-}$ and WT mice. $n = 6$ mice per group. Western blot strips (a target protein and a loading control) were cut from one gel. **d** Protein levels of Slc6a1, Slc6a11, and E4bp4 in the cortex of *Rev-erbα*$^{-/-}$ and WT mice. $n = 6$ mice per group. Western blot strips (a target protein and a loading control) were cut from one gel. **e** Effects of Rev-erbα overexpression or knockdown on mRNA expressions of *Slc6a1* and *Slc6a11* in Neuro-2a cells ($n = 3$ biologically independent samples). Two-sided *t* test *p* values (from left to right): 0.0008, 0.0004, 0.0149, and 0.0062. **f** Effects of Rev-erbα overexpression or knockdown on *Slc6a1* and *Slc6a11* mRNA expressions in primary hippocampus neurons ($n = 3$ biologically independent samples). Two-sided *t* test *p* values (from left to right): 0.0006, 0.0005, 0.0012, and 0.0004. **g** Effects of Rev-erbα overexpression or knockdown on mRNA expressions of *Slc6a1* and *Slc6a11* in primary cortex neurons ($n = 3$ biologically independent samples). Two-sided *t* test *p* values (from left to right): 0.0022, 0.0046, 0.0387, and 0.0266. **h** SR8278 dose-dependently represses mRNA expressions of *Slc6a1* and *Slc6a11* in primary hippocampus and cortex neurons ($n = 3$ biologically independent samples). *P* values (from left to right): 0.0065, 0.0016, 0.0024, 0.0016, 0.0341, 0.0065, 0.0019, 0.0004, 0.0499, 0.0193, 0.0089, 0.0046, 0.1162, 0.0118, 0.0033, and 0.0031 (one-way ANOVA and Bonferroni post hoc test). All data are presented as mean ± SEM. Statistics for **a–d** were performed with two-way ANOVA and Bonferroni post hoc test. * Represents a *p* value of <0.05. ZT zeitgeber time, Rel relative.

significant role in the regulation of circadian rhythms, inflammations, and metabolic homeostasis[36–38].

Our study suggests a potential mechanism for drug resistance in the treatment of epileptic seizures. Rev-erbα is a known activator of efflux transporters, including P-gp and Mrp2 (Fig. S19)[19,20]. Elevated Rev-erbα in the central nervous system (CNS) would drive expressions of those proteins in the brain–blood barrier that limit the entry of drug substrates into CNS, thereby causing drug resistance. This mechanism is supported by the facts that efflux transporters are overexpressed in endothelial cells from patients with refractory epilepsy and that some, but not all, AEDs such as tariquidar and lamotrigine are the substrates of P-gp and Mrp2[39,40]. We observed alleviated epileptic seizures in jet-lagged mice (a clock-disrupted model) owing to downregulated Rev-erbα. Interestingly, interference of clock system via bright light (called "light therapy") has been shown to be effective against epilepsy and related disorders[41,42]. However, this does not mean that physiological disruption of circadian clock is an appropriate measure to manage seizures because clockwork disturbance is associated with many other disorders (e.g., depression, cancers, metabolic disorders, and cardiovascular diseases) that may overwhelm the seizure problem[43]. An effective strategy should specifically modulate Rev-erbα expression or activity without interfering with the entire clock function.

Our findings support a critical role of GABA transporters (responsible for reuptake of GABA after synaptic release) in the pathophysiology of epilepsy, as proposed in the literature[6,9]. Rev-erbα-driven GABA transporters (Slc6a1 and Slc6a11) enhance GABA reuptake (clearance) and shorten IPSPs, thereby attenuating GABAergic inhibition and promoting epileptic seizures (Fig. 6). We may exclude the possibility of additional mechanisms, such as inflammation and synaptic phagocytosis contributing to Rev-erbα promotion of epileptic seizures because Rev-erbα is a repressor of glial activation, neuroinflammation, and synaptic phagocytosis[44,45]. The AED tiagabine acts directly on GABA transporters and inhibits their functions[9]. Contrasting with this, the antiepileptic agent SR8278 indirectly represses the expressions of GABA transporters via antagonizing Rev-erbα activity. Compared with GABA transporters, targeting Rev-erbα may have additional beneficial effects, including anti-Alzheimer's disease, enhanced cholesterol metabolism, and improved homocysteine homeostasis and ammonia clearance[36,46,47]. However, it is not without problems since inhibition of Rev-erbα may elicit detrimental effects on memory and mood[48,49]. Therefore, correcting Rev-erbα expression and activity to a normal range should be the therapeutic aim for epileptic seizures, while avoiding side effects potentially generated by a deficiency in Rev-erbα function.

In a prior report, Bmal1-deficient mice are seizure-sensitive based on measurements of electrical seizure thresholds, although the underlying mechanisms were unexplored[17]. This may suggest a protective effect of Bmal1 on epileptic seizures. The distinct roles of Bmal1 and Rev-erbα in regulating seizure susceptibility and other disorders [e.g., hepatitis C virus (HCV) infection] conform to their feedback loop association, which is known to be essential for generation of circadian rhythms[50,51]. On the other hand, the protective role of Bmal1 may suggest the involvement of alternative mechanisms independent of the Rev-erbα mechanism in regulating epileptic seizures considering Bmal1 is a transcriptional activator of Rev-erbα. This is possible because Bmal1 and Rev-erbα are showed to regulate biological processes via distinct pathways. For instance, Bmal1 regulates HCV replication through modulating viral receptors CD81 and claudin-1, whereas Rev-erbα regulates HCV replication via modulating SCD and subsequent release of infectious particles[51].

In summary, Rev-erbα is identified as a potential epileptogenic factor and a main source of seizure rhythmicity. Mechanistically, Rev-erbα drives the expressions of GABA transporters (Slc6a1 and Slc6a11) via transcriptional repression of E4bp4 (a negative regulator of the two transporters) and enhances GABA reuptake, thereby alleviating GABA-mediated inhibition and promoting epileptic seizures. These findings enhance our understanding of the crosstalk between circadian clock and epileptic seizures. Targeting Rev-erbα may represent a promising mechanism for the management of epileptic seizures.

## Methods

**Materials**. SR8278 was purchased from Tocris Bioscience (Ellisville, MO). NO-711, SNAP-5114, SR95531, kynurenic acid, and GABA-d6 were obtained from Sigma (St Louis, MO). Slc6a1 luciferase reporters (−2000/+37 bp and a mutated version), Slc6a11 luciferase reporters (−2500/+34 bp and a mutated version), pRL-TK, pcDNA3.1, pcDNA3.1-Rev-erbα, pcDNA3.1-E4bp4, siRev-erbα (siRNA targeting *Rev-erbα*), siE4bp4 (siRNA targeting *E4bp4*), and a negative control for siRNAs (siNC) were obtained from Transheep Technologies (Shanghai, China).

**Human specimens**. Surgical specimens (hippocampus and temporal cortex) were collected from ten patients with TLE and five patients with glioma (demographic information provided in Supplementary Table 2; time of sample collection: 14:00–16:00). Epileptogenic foci of TLE patients were identified by brain PET-CT, MRI scan, and EEG (Fig. S1). Glioma patients without seizure occurrence served as a control. The TLE patients underwent temporal lobectomy, whereas patients with glioma underwent tumor resection. Tissues were collected directly from the operating room, within 5 min of removal of the tissues from the blood supply. Informed consent was obtained from the patients (a template of informed consent form is provided in the Source data file). The content of informed consent includes disclosure of information, potential risks and benefits, and participants' rights. No tissue was ever removed solely for research purposes. The study protocol was approved by the institutional review board of the First Affiliated Hospital of Jinan University (KY-2020-039). All human studies complied with all relevant ethical regulations.

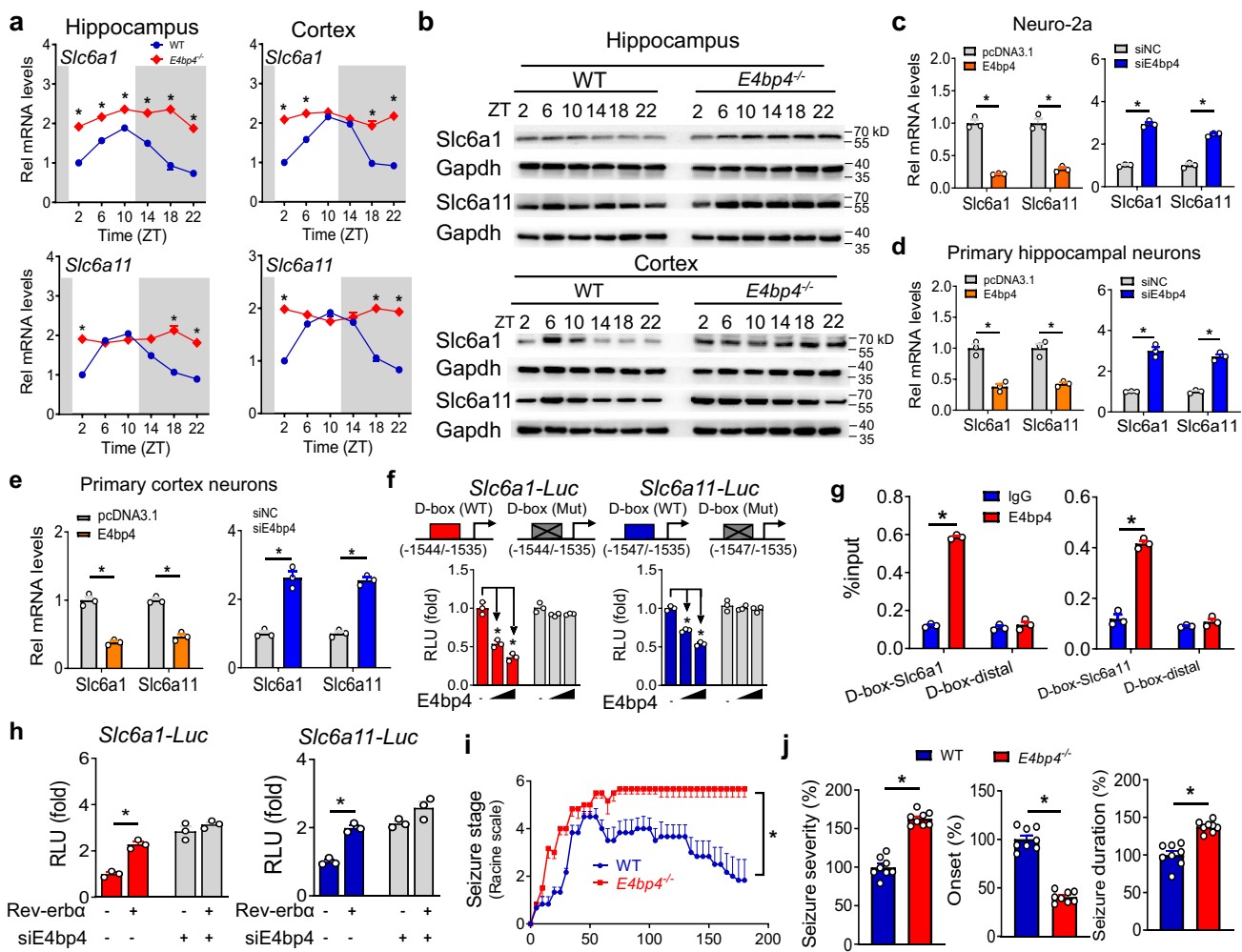

**Fig. 7 Rev-erbα regulates Slc6a1 and Slc6a11 expressions via repression of E4bp4. a** mRNA expressions of *Slc6a1* and *Slc6a11* in hippocampus and cortex of *E4bp4⁻/⁻* and wild-type (WT) mice. $n = 6$ mice per group. *$p < 0.05$ (two-way ANOVA and Bonferroni post hoc test). **b** Protein levels of Slc6a1 and Slc6a11 in hippocampus and cortex of *E4bp4⁻/⁻* and WT mice. $n = 6$ mice per group. Western blot strips (a target protein and a loading control) were cut from one gel. **c** Effects of E4bp4 overexpression or knockdown on mRNA expressions of *Slc6a1* and *Slc6a11* in Neuro-2a cells ($n = 3$ biologically independent samples). Two-sided *t* test *p* values: 0.0001, 0.0007, 0.0001, and 0.0001. **d** Effects of E4bp4 overexpression or knockdown on mRNA expressions of *Slc6a1* and *Slc6a11* in primary hippocampus neurons ($n = 3$ biologically independent samples). Two-sided *t* test *p* values: 0.0014, 0.0018, 0.0004, and 0.0002. **e** Effects of E4bp4 overexpression or knockdown on mRNA expressions of Slc6a1 and Slc6a11 in primary cortex neurons ($n = 3$ biologically independent samples). Two-sided *t* test *p* values: 0.0004, 0.0003, 0.001, and 0.0001. **f** Effects of E4bp4 on *Slc6a1* and *Slc6a11* promoter activities ($n = 3$ biologically independent samples). *P* values: 0.0002, 0.0001, 0.0033, and 0.0001 (one-way ANOVA and Bonferroni post hoc test). **g** ChIP assays showing recruitment of E4bp4 protein to *Slc6a1* and *Slc6a11* promoters ($n = 3$ biologically independent samples). Two-sided *t* test *p* values: 0.0001, 0.4384, 0.0001, and 0.2098. **h** Knockdown of *E4bp4* attenuates the activation effects of Rev-erbα on *Slc6a1* and *Slc6a11* promoter activities ($n = 3$ biologically independent samples). Two-sided *t* test *p* values: 0.0011 and 0.0003. **i** Seizure stages of *E4bp4⁻/⁻* and WT mice after injection of kainic acid at ZT6. $n = 6$ mice per group. $P = 0.0008$ (two-sided Kruskal–Wallis test). **j** Seizure parameters (severity, onset, and duration) of *E4bp4⁻/⁻* and WT mice after injection of kainic acid at ZT6. $n = 6$ mice per group. Two-sided *t* test *p* values: <0.0001, <0.0001, and <0.0001. All data are mean ± SEM. * Represents a *p* value of <0.05. Rel relative, Mut mutation.

**Mice**. Wild-type C57BL/6 mice were obtained from HFK Bioscience (Beijing, China). *Rev-erbα⁻/⁻* and *Rev-erbβ⁻/⁻* (C57BL/6 background) mice have been established and validated in our laboratory[36]. PCR genotyping information for *Rev-erbα* was as follows: a 516 bp fragment was indicative of wild-type allele, and a 920 bp fragment was indicative of null allele (Fig. S20). Heterozygotes showed the fragments of both 516 and 920 bp (Fig. S20). *E4bp4⁻/⁻* (C57BL/6 background) mice were obtained from Dr. Masato Kubo at RIKEN institute in Japan[52]. These knockout mice and wild-type (+/+) littermates were reproduced by intercrossing the heterozygous mice. All mice were maintained under a 12 h light/12 h dark cycle (at controlled room temperature of 22–25 °C and a relative humidity of 40–60%), and with free access to food and water. Male mice (8–12 weeks) were used for experiments. The animal experiments were approved by the Ethics Committee of Jinan University and the experimental procedures strictly followed the guidelines for Institutional Animal Care and Use.

**KA-induced acute seizure model**. Knockout (*Rev-erbα⁻/⁻*, *Rev-erbβ⁻/⁻*, and *E4bp4⁻/⁻*) mice and wild-type littermates were treated with KA (20 mg/kg, i.p., injected at ZT6 or ZT18) to induced acute seizures (status epilepticus or SE). SE was a continuous seizure activity that lasted for >10 min and also the different types of seizures (stages 1–5) were recurring at very short intervals (<1 min) during the SE. The stages in the SE were identified and recorded according to the Racine scale as previously described[53]: 0 (no response), 1 (staring and reduced locomotion), 2 (head nodding), 3 (unilateral forelimb clonus), 4 (bilateral forelimb clonus), 5 (rearing and falling), and 6 (status epilepticus and death). In addition to recording seizure scores, seizure severity was determined by integrating individual scores per mouse over the duration of the experiment. Twenty-four hours post KA treatment, mice were sacrificed to collect cortex and hippocampus.

In order to collect EEG data, mice were subjected to electrode implantation prior to acute seizure inducement. In brief, mice were anesthetized and mounted in a stereotaxic apparatus (RWD Life Science, Shenzhen, China). One electrode was

placed over the cerebellum to serve as a reference electrode and one electrode over the frontal cortex to serve as a background electrode, together with a third electrode implanted onto the temporal cortex (AP -2.3 mm; L -1.5 mm) for EEG recordings. Animals were allowed to recover for 7 days, and then treated with KA (20 mg/kg, i. p.). For monitoring, a pinnacle preamplifier was connected to the headmount and a swivel that was attached to the digitizer. Mice were subjected to 6 h EEG monitoring session consisting of 3 h baseline measurements and 3 h measurements post KA injection. Data were acquired using the PAL 8200 software (Pinnacle Technology) at a sample rate of 400 Hz. EEG signals were amplified with a band-pass filter setting of 0.5–100 Hz. Electrographic seizures were defined as spikes or sharp-wave discharges with amplitudes at least two times higher than baseline and lasting >5 s.

Additional set of experiments were performed to assess the effects of Rev-erbα function on epileptic susceptibility. Wild-type mice were pretreated with a Rev-erbα ligand (25 mg/kg SR8278 or 50 mg/kg SR9009, i.p., once daily, injected at ZT6) for 1 week[35,54]. Rev-erbα ligands were formulated in 10% DMSO:10% cremophor:80% phosphate-buffered saline (PBS) vehicle. Control mice were pretreated with vehicle. On day 8, all mice were treated with KA (20 mg/kg, i.p., injected at ZT6) to induced acute seizures. Seizures were scored as described above.

**Pilocarpine-induced chronic seizure model**. Rev-erbα[−/−] mice and wild-type littermates were subjected to electrode implantation as described in KA-induced acute seizure model. After recovery, mice were sequentially injected with scopolamine methyl nitrate (1 mg/kg, s.c.) and pilocarpine (300 mg/kg, i.p., injected at ZT6) with an interval of 30 min to induce status epilepticus. After 2 h, diazepam (10 mg/kg, i.p.) was administered to terminate seizures. Animals were subjected to EEG-video monitoring. To further assess Rev-erbα effects on epileptic seizures, wild-type model mice were treated with SR8278 (25 mg/kg, i.p., once daily, injected at ZT6) or SR9009 (50 mg/kg, i.p., once daily, injected at ZT6) or vehicle for 1 week.

**Rapid hippocampal kindling**. Rev-erbα[−/−] mice and wild-type littermates were anesthetized and mounted in a stereotaxic apparatus (RWD Life Science, Shenzhen, China). The electrodes (795500, A.M. Systems) were implanted into the right ventral hippocampus (AP -2.9 mm; L -3.0 mm; V -3.0 mm) for kindling stimulation and EEG recordings. An electrode was implanted onto the temporal cortex for EEG recordings. One screw was placed over the cerebellum to serve as a reference electrode and the other one over the frontal cortex to serve as a background electrode. After recovery for 7 days, the afterdischarge threshold (ADT) of each mouse was measured (monophasic square-wave pulses, 20 Hz, 1 ms/pulse, 40 pulses) with a constant-current stimulator (SEN-7203, Nihon Kohden) and EEGs were recorded with a Neuroscan system (NuAmps, Neuroscan System). The stimulation current was started at 60 μA and then increased by 20 μA every 5 min. The minimal current that produced at least a 5 s ADD was defined as the ADT of that mouse. Thereafter, mice received kindling stimulations (300 μA, monophasic square-wave pulses, 20 Hz, 1 ms/pulse, 40 pulses). Seizures were scored as described above. In addition, to assess the effects of Rev-erbα antagonism on epileptic susceptibility, wild-type mice were pretreated with the Rev-erbα antagonist SR8278 (25 mg/kg, i. p., once daily, injected at ZT6) or vehicle for 1 week. On day 8, mice received kindling stimulations (ZT6) to induced TLE. Seizures were scored as described above.

**RNA sequencing**. Brain tissue samples were collected from Rev-erbα[−/−] mice and wild-type littermates at ZT6 and ZT18. RNA was isolated using Trizol (Invitrogen, Carlsbad, CA) according to the manufacturer's instructions. RNA was quantified using Qubit™ 2.0 Fluorometer (Life Technologies, CA) and the quality was checked using Bioanalyzer 2100 RNA 6000 Nano Kit (Agilent Technologies, Santa Clara, CA). RNA samples were considered qualified when RIN > 7.5. 1 μg total RNA per sample was mixed with 2 μl 1:100 diluted ERCC RNA Spike-In Mix (Cat 4456740, Thermo Fisher Scientific, CA), followed by polyA mRNA selection for library preparation. Sequencing libraries were generated using NEBNext Ultra RNA Library Prep kit for Illumina (New England BioLabs, Ipswich, MA) following the manufacturer's recommendations, and index codes were added to attribute sequences to each sample. The indexed libraries were pooled from eight individual samples and then sequenced on Illumina HiSeq X Ten platform to generate 150 bp paired-end reads. Clean data (clean reads) were obtained by removing reads containing adapter and ploy-N using Fastp program, and aligned to mouse GRCm38/mm10 genome with HISAT2 v2.0.4 with default parameters as described[55]. Differentially expressed genes were identified using Cuffdiff v2.0.1 with default parameters[56]. Genes and transcripts were defined as differentially expressed if they showed a |fold change| > 1.5 and false discovery rate < 0.05.

**Pharmacokinetic analysis**. SR8278 (25 mg/kg) was administered to wild-type mice by intraperitoneal injection. Blood, brain, and liver samples were collected at 0.25, 0.5, 1, 2, 4, and 6 h (n = 5 per time point). Tissue samples were homogenized at a ratio of 1/2 (w/v) in saline solution. Plasma samples and tissue homogenates were subjected to deproteinization using acetonitrile. After vortex, the mixture was centrifuged, and the resulting supernatant was collected and dried using Eppendorf Concentrator Plus (Hamburg, Germany). The dry residuals were redissolved in

200-μl of acetonitrile/water (50:50, v/v). After centrifugation, a 5-μl aliquot of the supernatant was injected into the UPLC-QTOF/MS system (Waters, Milford, MA)[57].

KA (20 mg/kg) was administered to wild-type mice by intraperitoneal injection. Blood and brain (hippocampus and cortex) samples were collected at 0.25, 0.5, 1, and 2 h (n = 3 per time point). Samples were homogenized at a ratio of 1/2 (v/v or w/v) in 5% trichloroacetic acid (TCA), and centrifuged. Pellets were resuspended in 5% TCA and centrifuged again. Supernatants from the two centrifugation steps were combined and dried using Eppendorf Concentrator Plus (Hamburg, Germany). The dry residuals were redissolved and injected into the UPLC-QTOF/MS system (Waters, Milford, MA). KA in plasma and tissues were quantified by UPLC-QTOF/MS[58].

**Brain slice preparation**. Rev-erbα[−/−] mice and wild-type littermates were anesthetized and decapitated into an ice-cold slicing solution containing (in mM) 75 sucrose, 87 NaCl, 2.5 KCl, 0.5 CaCl$_2$, 4 MgCl$_2$, 24 NaHCO$_3$, 1.25 NaH$_2$PO$_4$, and 25 glucose. Brain slices (300-μm thick) were prepared in ice-cold slicing solution with a Leica VT-1200S Vibratome (Wetzlar, Germany). Slices containing hippocampus were incubated at 37 °C for 30 min in NMDG-HEPES recovery solution containing (in mM) 93 NMDG, 2.5 KCl, 0.5 CaCl$_2$, 93 HCl, 1.2 NaH$_2$PO$_4$, 10 MgSO$_4$, 30 NaHCO$_3$, 20 HEPES, 5 sodium ascorbate, 2 thiourea, 3 sodium pyruvate, and 25 glucose with an osmolarity of 310–320 mOsm/l, and then stored at room temperature for 1 h before use. All solutions were bubbled continuously with 95% O$_2$ and 5% CO$_2$.

**Whole-cell patch-clamp recordings**. Whole-cell patch-clamp recordings were performed with an IR-DIC visualization technique[59]. In brief, brain slices were transferred to a recording chamber and maintained at 32 ± 1 °C in artificial cerebrospinal fluid (aCSF) containing (in mM) 126 NaCl, 2.5 KCl, 1.25 NaH$_2$PO$_4$, 26 NaHCO$_3$, 2 MgCl$_2$, 2 CaCl$_2$, and 10 glucose. Data were collected with a Multiclamp 700B amplifier (Molecular Devices, Palo Alto, CA), low-pass filtered at 3 kHz and sampled at 10 kHz. Whole-cell recordings were obtained with patch pipettes (6–8 MΩ) filled with different internal solutions according to experiments. Cells were recorded with a holding potential of −70 mV. To monitor sIPSCs and tonic GABA currents, the glutamate receptor antagonist kynurenic acid (3 mM) were added to the perfusing aCSF. The internal solution contained (in mM) 10 K-gluconate, 125 KCl, 2 MgCl$_2$, 10 EGTA, 10 HEPES, 2 Mg$_2$ATP, 0.5 Na$_2$GTP, and 10 phosphocreatine. For mEPSC recordings, 1 μM TTX was added to the bath. The internal solution contained (in mM) 126 K-gluconate, 4 KCl, 10 HEPES, 4 Mg$_2$ATP, 0.3 Na$_2$GTP, and 10 phosphocreatine. sIPSC and mEPSC data were analyzed with the Clampfit software (Molecular Devices, Sunnyvale, CA). To assess tonic GABA currents, all-point histograms of amplitude values were generated from 20-s segments before and during SR95531 (a specific GABA$_A$ receptor antagonist, 10 μM) application, and fitted by Gaussian distribution with Origin software 8.0 (Northampton, MA).

**Isolation of primary hippocampus and cortex neurons**. Eighteen-day-old embryos from mice were dissected in Ca$^{2+}$ and Mg$^{2+}$ free Hanks' balanced salt solution (CMF-HBSS). After removal of meninges, hippocampus, and cortices were washed with CMF-HBSS, and the cortices were cut into small pieces. Hippocampus and cortex samples were incubated with 0.25% trypsin in CMF-HBSS for 20 min at 37 °C followed by incubation with 10% horse serum and DNaseI. The dissociated cells were resuspended and plated in minimum essential medium supplemented with 10% horse serum, 0.6% glucose, 2 mM glutamine, and 1 mM sodium pyruvate (Gibco). Three hours later, the medium was changed to neurobasal medium supplemented with B-27, 2 mM glutamine, 100 U/ml penicillin, and 100 μg/ml streptomycin (Gibco). Half of the culture medium was replaced with fresh medium every 3 days. Cells were cultured 12 days prior to experiments.

**Cell transfection and treatment**. Neuro-2a cells were cultured in MEM supplemented with 10% fetal bovine serum, 100 U/ml penicillin, and 100 μg/ml streptomycin (Gibco). Primary hippocampus and cortex neurons were cultured in neurobasal medium supplemented with B-27, 2 mM glutamine, 0.6% glucose, 100 U/ml penicillin, and 100 μg/ml streptomycin (Gibco). Cells were transfected with overexpression plasmids or siRNA (Supplementary Table 3) or controls using jetPRIME™ transfection reagent (Polyplus transfection, Illkirch, France). Four different sets of siRNAs were designed for each gene (Rev-erbα and E4bp4), and tested for their relative efficiency in gene silencing (Fig. S21). The most efficient one was used in knockdown experiments. The siRNA specificity was verified by rescue experiments (Fig. S21). Primary hippocampus and cortex neurons were treated with different concentrations of SR8278 or vehicle. Twenty-four hours after transfection or treatment, the cells were collected for qPCR assays.

**GABA uptake experiments**. In the first set of uptake experiments, primary hippocampus and cortex neurons were transfected with siRev-erbα or siNC. After 36 h, cells were preincubated in HBSS for 15 min, and then incubated with HBSS containing a serial concentration of GABA-d6. After another 15 min, cells were collected and GABA-d6 was quantified by UPLC-QTOF/MS. In the second set of uptake experiments, sagittal brain slices containing hippocampus were placed into 12-well plates. After preincubation with NMDG-HEPES recovery solution for

30 min at 37 °C, slices were incubated with GABA-d6 solution (at a series of concentrations). Uptake transport was terminated at 10 min by adding 1 ml ice-cold methanol. The amounts of GABA-d6 taken up by neurons were determined by UPLC-QTOF/MS.

GABA-d6 quantification was performed using Waters ACQUITY UPLC-Xevo G2 QTOF/MS system with a BEH Amide column (1.7 μm, 2.1 × 100 mm; Waters, Milford, MA). Parameters for mass spectrometer were set as previously described[36]. Gradient elution was applied with 20 mM ammonium acetate in water as mobile phase A and 0.1% formic acid in acetonitrile as mobile phase B. The gradient program consisted of 0–3 min, 85% B; 3–7 min, 85–70% B; 7–12 min, 70–50% B; 12–16 min, 50–5% B; 16–20 min, 5–85% B. The flow rate was 0.3 ml/min.

**qRT-PCR**. Total RNA was isolated from cell and tissue samples using RNAiso plus (Takara, Shiga, Japan), and reversely transcribed to cDNA using Hiscript II Q RT SuperMix (Vazyme, Nanjing, China). qPCR reaction was performed following our published procedures[46]. *Cyclophilin b* or *GAPDH* was used as an internal control. Relative mRNA levels were determined using the $2^{-\Delta\Delta CT}$ method. Primer sequences are summarized in Supplementary Table 4.

**Western blotting**. Protein samples were separated by 10% SDS–PAGE and electrotransferred onto PVDF membranes. After blocking, the membranes were probed with primary antibodies (anti-Rev-erbα, anti-Slc6a1, anti-Slc6a11, anti-Bmal1, anti-Dbp, anti-Clock, and anti-E4bp4), followed by incubation with secondary antibody. Antibody information is provided in Supplementary Table 5. After adding enhanced chemiluminescence, the blots were imaged using an Omega Lum™ G imaging system (Aplegen, Pleasanton, CA), and quantified using the Quantity One software (Bio-Rad, Hercules, CA). Gapdh/GAPDH was used an internal control.

**Immunohistochemistry**. Brain samples were fixed in 4% paraformaldehyde, and then transferred to a grade series of sucrose solution (10, 20, and 30%). Coronal brain sections (a thickness of 20 μm) were blocked with 10% horse serum and 0.5% Triton X-100 in PBS, and then incubated with antibodies to Rev-erbα, NeuN, GFAP, or Iab1 (Supplementary Table 5). After washing with PBS, sections were incubated with secondary antibodies and DAPI. Sections were washed, mounted, and imaged using Zeiss LSM780 confocal microscope (Jena, Germany; for Fig. 1c–e) or Nikon Optiphot fluorescent microscope (Tokyo, Japan; for Figs. 2f and 3f). ImageJ software was used to count Rev-erbα-, NeuN-, GFAP-, and Iab1-positive cells. At least three regions from each section and three sections were imaged for each animal, and three animals were used for each group.

**FJB and TUNEL staining**. To detect neurodegeneration, coronal sections from mouse hippocampus were stained using FJB (Millipore). Irreversible DNA damage was assessed using a fluorescein-based TUNEL kit (Roche). Images of staining were captured using a Nikon Optiphot fluorescent microscope (Tokyo, Japan). ImageJ software was used to count FJB- and TUNEL-positive cells. At least three regions from each section and three sections were imaged for each animal, and three animals were used for each staining.

**Luciferase reporter assay**. Cells were co-transfected with luciferase reporter, pRL-TK, and expression plasmid using jetPRIME™ transfection reagent (Polyplus Transfection, Illkirch, France). After 24 h transfection, cells were lysed in a lysis buffer, and the luciferase activities were measured using a Dual-Luciferase® Reporter Assay System (Promega, Madison, WI). Firefly luciferase activity was normalized to renilla luciferase activity, and expressed as relative luciferase unit.

**ChIP assay**. ChIP assays were performed using a SimpleChIP Enzymatic Chromatin IP Kit (Cell Signaling Technology, Beverly, MA), according to the manufacturer's instructions. Mouse brain tissues were fixed in 1% formaldehyde and lysed with lysis buffer, followed by digestion with micrococcal nuclease. Sheared chromatin was immunoprecipitated with anti-E4bp4 antibody or normal IgG (a negative control) for overnight at 4 °C. The immune complex was decross-linked at 65 °C for 4 h. The obtained DNAs were purified and analyzed by qPCR with specific primers (Supplementary Table 3).

**Statistical analysis**. Data are presented as raw data (mean ± SEM) and tested for normality with a Shapiro–Wilk test, outliers were included in data analysis. ANOVA (one-way or two-way) with Bonferroni post hoc test or Kruskal–Wallis test was used for multiple comparisons (GraphPad Prism, San Diego, CA). Two group comparisons were performed with Student's *t* test or nonparametric Mann–Whitney test. *P* values <0.05 were considered statistically significant.

**Reporting summary**. Further information on research design is available in the Nature Research Reporting Summary linked to this article.

## Data availability

RNA-seq data have been deposited to Sequence Read Archive (SRA) under the accession number PRJNA637449. Other data supporting the findings of this study are available within the paper and its Supplementary Information files. Source data are provided with this paper.

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

## Acknowledgements

This work was supported by the National Natural Science Foundation of China (Grant 81722049, 81671288, and 81903698). We sincerely thank Dr. Zhigang Wang (the First Affiliated Hospital of Jinan University) and Dr. Lei Shi (College of Pharmacy, Jinan University) for their technical expertise and support.

## Author contributions

T.Z. and B.W. conceived and designed the study. T.Z., F.Y., H.X., M.C., X.C., L.G., C.Z., Y.X., J.Y., and F.W. conducted experiments. T.Z., F.Y., H.X., and J.Y. performed data analysis. T.Z. and B.W. wrote and revised the manuscript.

## Competing interests

The authors declare no competing interests.
