## [Peer Review File · Nature Communications]

Reviewers' Comments:

Reviewer #1:

Remarks to the Author:

This paper establishes a connection between increased REV-ERBa activity and epilepsy. The authors first show increased Rev-erba in both human epilepsy tissue and in the mouse kainic acid model of epilepsy. Next, they tested the effect of REV-ERBa deletion on epilepsy. The REV-ERBa ^{-/-} mice showed improvement in multiple seizure parameters including severity and onset time. Use of a small molecule to inhibit REV-ERBa resulted in a similar amelioration of the seizure phenotype. When REV-ERBa was deleted or inhibited in a hippocampal kindling model of epilepsy, they experienced less severe seizures, shorter after-discharge duration, and increased stimulations required for a generalized seizure. Whole cell patch clamp electrophysiology demonstrated changes in inhibition. By comparing transcriptomes of WT and REV-ERBa KO mice they found dysregulation of SLC6a1 and SLC6a11, transporters responsible for GABA reuptake. Knockout and knockdown of REV-ERBa confirmed that the GABA concentration is higher in WT neurons and brain slices. By knocking out or overexpressing REV-ERBa, neurons were found to have decreased and increased levels (respectively) of both transporters in a dose-dependent manner. REV-ERBa was found to be responsible for increasing the transporters through its inhibition of E4BP4 transcription.

This study is an important addition to how alteration of component of the molecular clock is important for epilepsy. Although it was previously known that knockout of CLOCK worsened the epilepsy phenotype, the dynamics of the circadian molecular clock were not well defined. The researchers convincingly showed how REV-ERBa levels change at different times of day and how the knockout affected epilepsy at those specific time points. To gather this evidence required thoroughness and acquiring samples from ZT2-ZT22 revealed characteristics of the knockout that would have been missed by using a more simplistic window. Throughout this paper the authors routinely showed both qPCR and Western Blot data to reinforce their observations, leaving no doubt about regulation on a transcriptional or translational level. Establishing the relationship between REV-ERBa and epilepsy through knockout and small molecule ablation was a significant finding but diving deeper into the direct effectors using a screen increased the impact of the work. To further increase the clarity of this pathway the researchers identified the trans-deactivating component E4bp4 and even discovered the specific genomic binding domain E4bp4 targeted. While much of the study is well performed, there are some issues in the presentation of the data. The investigators make the case that knockout of REV-ERBa decreases some of the negative characteristics from acute kainic acid treatment. While gliosis may be evident, the Fluor Jade staining is not particularly convincing and could use clarity, especially concerning the specific area of the hippocampus. Quantification would also be useful here. The NeuN staining also does not show a convincing change in the neuron number. Quantification would be necessary to concretely prove a change in cell number. The electrophysiology presented later in the paper, namely in Figure 4, is compelling. But a power spectrum analysis would be helpful in determining a significant difference as opposed to merely observation. In this same figure, numbers of stimulations for a generalized seizure are compared between WT and KO (4C and G). Is there any explanation as to why the WT animals vary so much between both groups? It appears to have doubled from ~12 to ~25 stimulations. Additionally, siRNAs are used throughout the paper but there does not seem to be data showing the use of multiple siRNAs or rescue to ensure specificity. There also does not appear to be a section on the sequences or applications of the siRNAs in the methods section.

Minor points/ typos

- P3 line 16: change to "into presynaptic neurons"
- P4 line 1: change to "therefore enhancing GABAergic function"
- P4 line 17: should "activation" be "overexpression"?
- P6 line 3: should "specimens" be "samples"?
- P10 line 15: change "neuron" to "neurons"
- P10 line 16: "(clearance) consistent" should be "(clearance) is consistent"

- P12 line 4: change "in gene promoter" to "in the gene's promoter"
- P12 line 17: change "in gene promoters" to "in their promoters"
- P14 line 19: change "hippocampus neurons" to "hippocampal neurons"
- P15 line 5: change "they may do not contribute" to "they may not contribute"
- P15 line 6: change "Tef is mildly fluctuated" to "Tef fluctuates mildly"
- P15 line 7: entails should maybe be replaced by requires
- For all figures and methods: Change "GAFF" to GFAP" and "DIPA" to "DAPI"
- P20 line 18: Did you coinject scopolamine with the pilocarpine or did you wait after the scopolamine injection?
- Fig 2 and 3: Put DAPI images before other images

Reviewer #2:

Remarks to the Author:

Methods: For studies involving KO mice, more details are required on the wild type controls including their source and genetic background.

Methods: p.18 lines 8-9: There are two separate sources given for Slc6a1 and no sources given for Slc6a11. It is possible that one of the Slc6a1 entries is a typographical error.

Methods: Western blot and IHC experiments require further detail especially with regards to quantification and controls. More details are required on the nature of the antibodies, both primary and secondary. Specific product numbers should be provided for primary antibodies.

Methods: The description of RNAseq methods is sparse and it is difficult to evaluate whether they are sufficiently rigorous particularly because no references are provided. Therefore, the following should be addressed:

- 1) What was the RNA Integrity Number (RIN) cutoff for inclusion of an RNA sample in library preparation? What kind of RNA Chip did they use for the Bioanalyzer?
- 2) With regard to library construction: Were manufacturer recommendations followed? What was the starting amount of RNA input into library preparation? Was it total RNA, polyA purified RNA or ribo-depleted RNA?
- 3) Was there a spike-in decoy used for NGS sequencing? If so, what concentration? Also, if so, how were they removed before alignment? Were barcodes used for the RNA library preparations? If so, how many samples were pooled and multiplexed for NGS sequencing?
- 4) What was the format of the paired end reads? 2x100bp, 2x150bp, etc.? What program was used to remove flow cell adapter and sequencing primer portions of the reads before alignment with HISAT2? Options were used for the alignment could be provided. Similarly, for Cuttdiff, analysis parameters could be given.

Methods: Primer sequences for gene knockdown are given as supplementary information but this is not referred to in the body of the manuscript.

Methods Supplementary Table 1: typographical errors: "women" should be "female" and "men" should be "male"

Methods p.19 line 15: How was status epilepticus defined?

Methods p.20 lines 14-15: How was the dose of the Rev-Erba ligands chosen? What were the vehicles? Provide a reference documenting their mechanism and specificity.

Methods: Chemoconvulsant studies are limited by use of a single drug dose. Generating and comparing dose-response curves provides the best assessment differences in seizure susceptibility between two strains or lines of mice. This criticism is tempered by use of different seizure testing paradigms involving two mechanistically distinct agents as well as a non-pharmacological model

involving electrical kindling. However, the authors should specify the time-of-day when seizure tests were performed, when the seizure stimulus was administered or when tissue samples were collected for each experiment.

Results: Summary results should be presented for fluoro-jade, TUNEL and other IHC experiments. It is not possible to evaluate the strength of the Fig.3 D-F experiments based only on single representative data points. Further, these studies were performed at a relatively short time point (24 hours) after a single 20 mg/kg i.p. dose of KA. The authors should refer to other studies in which such a treatment paradigm produced neuropathological changes and state whether their results are consistent. Numerous studies report that C57BL/6 mice are particularly resistant to the effects of KA.

Results p.8 lines 5-7 It is stated that Rev-ErbB is not likely involved in seizures given the supplemental data shown in Fig.5. However, that figure shows the results of a single seizure paradigm (acute seizures) where only 1 dose of KA was tested. Thus, the statement on p.8 is not completely justified. Similarly, the statements on p.15 lines 13-17 also require modification in light of this criticism.

Results: What other genes does Rev-Erba regulate besides E4BP4 that could impact seizure susceptibility? What other genes besides the Slc6a1 and Slc6a11 does E4bp4 regulate which could also impact seizure susceptibility?

Results: Were EAATs evaluated? If not, the premise that the role of glutamate in this model can be excluded (e.g. p.11 line 8) is not justified.

Results/Discussion: It is not clear whether the changes in expression are occurring in glia as well as neurons. If also in glia, a role for alternative mechanisms such as their role in inflammation or synaptic function must be considered.

Discussion p.16 lines 2-4: This section misses the point that AED transport by p-gp is a controversial topic and to suggest that the Rev-erb ligand regulates AED transport via p-gp requires some data or reference. Thus, this statement should at least be modified to indicate that there are AEDs that are clearly not p-gp substrates and p-gp has not been shown definitively to play a major role in AED resistance.

Discussion: Prior studies showed that Bmal1 is a transcriptional activator for Rev-erba, and Bmal1 KO mice are seizure-sensitive as well as arrhythmic regarding seizure threshold. More discussion of the interplay between these two molecules is warranted.

Reviewer #3:

Remarks to the Author:

This study examines the expression of the CLOCK component Rev-erbalpha in human tissue resected from patients with temporal lobe epilepsy, and then attempts to establish a connection between this altered expression and seizure propensity in multiple mouse models of seizures, using both transgenic and small molecule agonist and antagonist approaches. The authors conclude that circadian changes in Rev-erbalpha expression may influence seizures by altering GABA uptake, with associated changes in inhibitory synaptic transmission. In general, I find the manuscript to be interesting and, for the most part, competently conducted. I have several suggestions for revision:

The study is very ambitious and uses multiple techniques in a complicated experimental approach, including in vivo and in vitro physiology, RNA and protein profiling, transgenic and small molecule drug Rev-erbalpha manipulations, tissue culture, multiple seizure models, etc. This ambition is

both a strength and a weakness of the MS, as the breadth of the approach precludes in depth analyses and follow-up on important findings.

The information concerning Rev-erbalpha expression in human tissue resected from patients with temporal lobe epilepsy is critical to the premise of the whole manuscript, and one of the few direct putative linkers between the CLOCK system and TLE. As presented, there are serious issues with interpretation of this RNA and protein expression data. As is common and depicted in the patient data table (supp. Table 1), it frequently takes decades between initial emergence of seizures and surgical resection, during which time extensive pathology builds in the affected tissues, including extensive neuronal loss, gliosis, inflammation, and other changes. Loss of entire neuronal components of the hippocampus is not uncommon. In one study cited by the authors (Li et al., 2017), CLOCK is expressed only in neurons (and not in GFAP positive astrocytes) in human brain. Several studies have also implicated inflammation and CLOCK gene function as strongly interrelated. Given that the neuronal contribution to the tissue harvested for RNA and protein profiling undoubtedly had a reduced neuronal component relative to astrocytes, that there was enhanced inflammation in the tissue, and that these changes were restricted to the TLE (and not the brain tumor) tissue, it becomes very hard to interpret the data depicted in Fig. 1A–D without cell specific profiling.

The systemic KA model utilized in much of the manuscript generates data that is also hard to interpret. This is unfortunate since the primary putative linkage between circadian rhythmicity and seizures is derived from studies using this manipulation. Studies like those in the present MS usually profile seizure thresholds to assess pro- (or anti-) seizure effects of experimental manipulations. Injecting a suprathreshold systemic convulsant, and then profiling the seizure stage ends up convolving numerous contributing factors (e.g. drug metabolism, BBB permeability, seizure generalization mechanisms, seizure propensity, mortality, etc.) into the analysis, any of which could contribute to altering seizure stage.

The most applicable TLE model utilized in this study was the pilocarpine model of spontaneous seizures, depicted in SFig 8. This model generates mice with spontaneous seizures which involve the hippocampus, and a pathological profile which recapitulates important aspects of human TLE (mesial temporal sclerosis). Broadening the assessment of Rev-erbalpha effects on this model would improve the MS. However, some aspects of the spontaneous seizure data are curious. Pilocarpine mice have spontaneous seizures in 9-10 day clusters, separated by a week or more during which there are few or no seizures. This seizure biology increases the variability in aggregate seizure plots like those in SFig 8 greatly (since mice do not usually synchronize across populations), and often necessitates depiction of data in a more accessible format, i.e. individual mouse data for the two populations. The small error bars depicted in this figure are hard to reconcile with this typical manifestation of seizures in this mouse model.

The sIPSC effects in Fig. 5 are fairly subtle, and the frequency effects are hard to interpret. If in fact increased GABA uptake is the explanation for the sIPSC changes, then I would expect an enhancement in sIPSC decay time (albeit a more pronounced change than the few % change depicted), but the frequency effects are difficult to interpret. This suggests enhanced interneuron firing, but due to what? Perhaps augmenting this data with recordings of tonic GABA currents would enhance support for the putative linkage between transporter expression and inhibitory function. As it stands, the subtle effects described on sIPSCs seems somewhat at odds with the much greater seizure effects described in the remainder of the MS. In addition, these effects differ at least in part from the Li et al., 2017 findings when CLOCK was deleted in glutamatergic neurons, where sIPSC effects were much more pronounced and included amplitude, rise and decay time, and frequency changes. This should at least be discussed. In addition, physiological studies should provide data about cellular properties such as input resistance, membrane potential, firing properties, etc. that are routinely recorded during these types of studies. This could be included as supplemental data, but its absence is an issue.

Supplemental Figure 3 is not informative. Perhaps supplemental videos would improve presentation of the finding.

I did not understand the utility of the jet lag data (supplemental figure 4) and how this contributed to the overall development of the MS.

We wish to thank the reviewers for careful and valuable reviews. Below are our point-by-point responses.

Responses to reviewer #1

This paper establishes a connection between increased REV-ERB α activity and epilepsy. The authors first show increased Rev-erb α in both human epilepsy tissue and in the mouse kainic acid model of epilepsy. Next, they tested the effect of REV-ERB α deletion on epilepsy. The REV-ERB α $-/-$ mice showed improvement in multiple seizure parameters including severity and onset time. Use of a small molecule to inhibit REV-ERB α resulted in a similar amelioration of the seizure phenotype. When REV-ERB α was deleted or inhibited in a hippocampal kindling model of epilepsy, they experienced less severe seizures, shorter after-discharge duration, and increased stimulations required for a generalized seizure. Whole cell patch clamp electrophysiology demonstrated changes in inhibition. By comparing transcriptomes of WT and REV-ERB α KO mice they found dysregulation of SLC6a1 and SLC6a11, transporters responsible for GABA reuptake. Knockout and knockdown of REV-ERB α confirmed that the GABA concentration is higher in WT neurons and brain slices. By knocking out or overexpressing REV-ERB α , neurons were found to have decreased and increased levels (respectively) of both transporters in a dose-dependent manner. REV-ERB α was found to be responsible for increasing the transporters through its inhibition of E4BP4 transcription.

This study is an important addition to how alteration of component of the molecular clock is important for epilepsy. Although it was previously known that knockout of CLOCK worsened the epilepsy phenotype, the dynamics of the circadian molecular clock were not well defined. The researchers convincingly showed how REV-ERB α levels change at different times of day and how the knockout affected epilepsy at those specific time points. To gather this evidence required thoroughness and acquiring samples from ZT2-ZT22 revealed characteristics of the knockout that would have been missed by using a more simplistic window. Throughout this paper the authors routinely showed both qPCR and Western Blot data to reinforce their observations, leaving no

doubt about regulation on a transcriptional or translational level. Establishing the relationship between REV-ERB α and epilepsy through knockout and small molecule ablation was a significant finding but diving deeper into the direct effectors using a screen increased the impact of the work. To further increase the clarity of this pathway the researchers identified the trans-deactivating component E4bp4 and even discovered the specific genomic binding domain E4bp4 targeted.

Response: Thanks very much for a long and nice summary of the key points of our work.

While much of the study is well performed, there are some issues in the presentation of the data. The investigators make the case that knockout of REV-ERB α decreases some of the negative characteristics from acute kainic acid treatment. While gliosis may be evident, the Fluorojade staining is not particularly convincing and could use clarity, especially concerning the specific area of the hippocampus. Quantification would also be useful here. The NeuN staining also does not show a convincing change in the neuron number. Quantification would be necessary to concretely prove a change in cell number.

Response: We have followed the reviewer's suggestion, and have provided the quantification data on Fluoro-jade and TUNEL staining (please see new supplementary Figure 3).

The electrophysiology presented later in the paper, namely in Figure 4, is compelling. But a power spectrum analysis would be helpful in determining a significant difference as opposed to merely observation.

Response: EEG power spectra have been provided in supplementary Figure 12.

In this same figure, numbers of stimulations for a generalized seizure are compared between WT and KO (4C and G). Is there any explanation as to why the WT animals vary so much between both groups? It appears to have doubled from ~12 to ~25 stimulations.

Response: We appreciate the reviewer's careful review. We have gone back to original data, and identified a mistake in data analysis of Figure 4G. We regret for this careless. The data have been fixed (please see new Figure 4G). There is indeed no significant difference in the numbers of stimulations for GS between the WT group in Figure 4C and the vehicle group in Figure 4G.

Additionally, siRNAs are used throughout the paper but there does not seem to be data showing the use of multiple siRNAs or rescue to ensure specificity. There also does not appear to be a section on the sequences or applications of the siRNAs in the methods section.

Response: In fact, four different sets of siRNAs were designed and synthesized for each gene (*Rev-erba* and *E4bp4*), and tested for their relative efficiency in gene silencing. Thereafter, the most efficient one was used in knockdown experiments. The efficiency testing data have been provided in supplementary Figure 21 and the sequences of siRNAs in supplementary Table 3. Following the reviewer's suggestion, we have performed new (rescue) experiments to verify the siRNA specificity. The results showed that siRNA effect can be reversed by gene overexpression, confirming specific action of the used siRNA. The new data have been added to supplementary Figure 21. We have also followed the reviewer's suggestion, and described the application of siRNAs in the methods section (under "*Cell transfection and treatment*") in revised manuscript.

Minor points/ typos

- P3 line 16: change to “into presynaptic neurons”
- P4 line 1: change to “therefore enhancing GABAergic function”

Response: Revised as suggested. Thanks.

- P4 line 17: should “activation” be “overexpression”?

Response: We believe that “activation” may be more appropriate because the relevant experiment refers to modulation of seizure sensitivity by SR9009, a small-molecule agonist of Rev-erba.

- P6 line 3: should “specimens” be “samples”?
- P10 line 15: change “neuron” to “neurons”
- P10 line 16: “(clearance) consistent” should be “(clearance) is consistent”
- P12 line 4: change “in gene promoter” to “in the gene’s promoter”
- P12 line 17: change “in gene promoters” to “in their promoters”
- P14 line 19: change “hippocampus neurons” to “hippocampal neurons”
- P15 line 5: change “they may do not contribute” to “they may not contribute”
- P15 line 6: change “Tef is mildly fluctuated” to “Tef fluctuates mildly”
- P15 line 7: entails should maybe be replaced by requires

Response: Revised as suggested. Thanks.

- For all figures and methods: Change “GAFP” to GFAP” and “DIPA” to “DAPI”

Response: Fixed. Thanks for pointing this out.

- P20 line 18: Did you coinject scopolamine with the pilocarpine or did you wait after the scopolamine injection?

Response: We regret that our statement was not clear. In fact, mice were sequentially injected with scopolamine and pilocarpine with an interval of 30 min. The previous statement has been changed to “Wild-type mice were sequentially injected with scopolamine methyl nitrate (1 mg/kg, s.c.) and pilocarpine (300 mg/kg, i.p., injected at ZT6) with an interval of 30 min to induce status epilepticus.”

- Fig 2 and 3: Put DAPI images before other images

Response: Revised as suggested. Thanks.

Responses to reviewer #2

Methods: For studies involving KO mice, more details are required on the wild type controls including their source and genetic background.

Response: Wild-type controls in studies involving KO mice are matched littermates (+/+ genotype). Both KO and wild-type littermates were reproduced by intercrossing the heterozygous mice. This has been specified in revised manuscript. Please note that all mice used in the present study were C57BL/6 background, as indicated in the methods section (under "*Mice*").

Methods: p.18 lines 8-9: There are two separate sources given for Slc6a1 and no sources given for Slc6a11. It is possible that one of the Slc6a1 entries is a typographical error.

Response: This typo has been fixed. Thanks.

Methods: Western blot and IHC experiments require further detail especially with regards to quantification and controls. More details are required on the nature of the antibodies, both primary and secondary. Specific product numbers should be provided for primary antibodies.

Response: We accept the reviewer's criticism that more details should be provided regarding Western blot and IHC experiments. Accordingly, the experimental details (the nature of antibodies, quantification method and/or controls) have been added to supplementary Table 5 and to the methods section (under "*Western blotting*" and "*Immunohistochemistry*").

Methods: The description of RNAseq methods is sparse and it is difficult to evaluate whether they are sufficiently rigorous particularly because no references are provided.

Therefore, the following should be addressed:

1) What was the RNA Integrity Number (RIN) cutoff for inclusion of an RNA sample in library preparation? What kind of RNA Chip did they use for the Bioanalyzer?

2) With regard to library construction: Were manufacturer recommendations followed? What was the starting amount of RNA input into library preparation? Was it total RNA, polyA purified RNA or ribo-depleted RNA?

3) Was there a spike-in decoy used for NGS sequencing? If so, what concentration? Also, if so, how were they removed before alignment? Were barcodes used for the RNA library preparations? If so, how many samples were pooled and multiplexed for NGS sequencing?

4) What was the format of the paired end reads? 2x100bp, 2x150bp, etc.? What program was used to remove flow cell adapter and sequencing primer portions of the reads before alignment with HISAT2? Options were used for the alignment could be provided. Similarly, for Cuttdiff, analysis parameters could be given.

Response: We accept the reviewer's criticism that some details were missing in the description of RNA-seq. Accordingly, we have re-written the "*RNA-sequencing*" part (shown below) in the METHODS section, that have included the details requested by the reviewer.

"Brain tissue samples were collected from Rev-erba^{-/-} mice and wild-type littermates at ZT6 and ZT18. RNA was isolated using Trizol (Invitrogen, Carlsbad, CA) according to the manufacturer's instructions. RNA was quantified using Qubit™ 2.0 Fluorometer (Life Technologies, CA) and the quality was checked using Bioanalyzer 2100 RNA 6000 Nano Kit (Agilent Technologies, Santa Clara, CA). RNA samples were considered qualified when RIN > 7.5. 1 µg total RNA per sample was mixed with 2 µl 1:100 diluted ERCC RNA Spike-In Mix (Cat 4456740, Thermo Fisher Scientific, CA), followed by

polyA mRNA selection for library preparation. Sequencing libraries were generated using NEBNext Ultra RNA Library Prep kit for Illumina (New England BioLabs, Ipswich, MA) following the manufacturer's recommendations and index codes were added to attribute sequences to each sample. The indexed libraries were pooled from eight individual samples and then sequenced on Illumina HiSeq X Ten platform to generate 150 bp paired-end reads. Clean data (clean reads) were obtained by removing reads containing adapter and ploy-N using Fastp program, and aligned to mouse GRCm38/mm10 genome with HISAT2 v2.0.4 with default parameters as described. Differentially expressed genes (DEGs) were identified using Cuffdiff v2.0.1 with default parameters. Genes and transcripts were defined as differentially expressed if they showed a |fold-change| > 1.5 and false discovery rate (FDR) < 0.05."

Please note that index codes (unique identifiers) instead of barcodes were used in our RNA-seq. Index codes and barcodes are two types of data tags. Index code is a characteristic DNA sequence used to distinguish and gather together similar items. Indexing approach enables parallel sequencing of different samples through the labeling of each library with a distinguishing index (Mol Ecol Resour. 2019; PMID: 30848092 // PLoS One. 2011; 6(10): e26426 // Nucleic Acids Res. 2012;40(1):e3 // BMC Genomics. 2016;17(1):876 // Methods. 2013;63(2):126-34)..

Methods: Primer sequences for gene knockdown are given as supplementary information but this is not referred to in the body of the manuscript.

Response: Fixed. Thanks.

Methods Supplementary Table 1: typographical errors: "women" should be "female" and "men" should be "male"

Response: Fixed. Thanks.

Methods p.19 line 15: How was status epilepticus defined?

Response: Status epilepticus is a continuous seizure lasting more than 5 min, or two or more seizures without full recovery of consciousness between any of them. We believe it may be unnecessary to define status epilepticus in the manuscript because it is a standard terminology in neurology.

Methods p.20 lines 14-15: How was the dose of the Rev-Erba ligands chosen? What were the vehicles? Provide a reference documenting their mechanism and specificity.

Response: We have cited two references (i.e., Nature. 2012;485(7396):62-8 // ACS Chem Biol. 2011;6(2):131-4) documenting the mechanisms and specificity of the used Rev-erba ligands in the revised manuscript. Please note that the dose was chosen according to the dose-response experiments in the literature [Nature. 2012;485(7396):62-8 (SR9009) // Mol Metab. 2017;6(7):703-714 (SR8278) // Cell. 2014;157(4):858-68 (SR8278) // Sci Rep. 2017;7(1):17142 (SR8278)]. Also, the vehicle (10% DMSO:10% cremophor:80% PBS) for Rev-erba ligands has been indicated in revised version. Please note that the vehicle was chosen based on the literature (Mol Metab. 2017;6(7):703-714).

Methods: Chemoconvulsant studies are limited by use of a single drug dose. Generating and comparing dose-response curves provides the best assessment differences in seizure susceptibility between two strains or lines of mice. This criticism is tempered by use of different seizure testing paradigms involving two mechanistically distinct agents as well as a non-pharmacological model involving electrical kindling.

However, the authors should specify the time-of-day when seizure tests were performed, when the seizure stimulus was administered or when tissue samples were collected for each experiment.

Response: We have followed the reviewer's suggestion, and have specified the time-of-day when seizure tests were performed, when the seizure stimulus was administered, and when tissue samples were collected for each experiment in revised manuscript.

Results: Summary results should be presented for fluoro-jade, TUNEL and other IHC experiments. It is not possible to evaluate the strength of the Fig.3 D-F experiments based only on single representative data points.

Response: We accept the reviewer's criticism. Accordingly, we have provided the quantification (summary) data on Fluoro-jade and TUNEL staining base on three independent experiments (please see new supplementary Figure 3).

Further, these studies were performed at a relatively short time point (24 hours) after a single 20 mg/kg i.p. dose of KA. The authors should refer to other studies in which such a treatment paradigm produced neuropathological changes and state whether their results are consistent. Numerous studies report that C57BL/6 mice are particularly resistant to the effects of KA.

Response: We understand the reviewer's concern about whether 24-h KA treatment can cause neuropathological changes. We did observe obvious loss of hippocampal neurons at 24-h after KA treatment (Figures 2 and 3). This is well consistent with previous reports by other investigators in which 24-h KA treatment induced neuronal loss in mouse hippocampus (Nat Med. 2012;18(7):1087-94 // Neuroscience.

2011;189(1-2):316-29 // Neuroscience. 2007;150(2):467-77). Therefore, it may be of little concern that 24-h KA treatment can cause neuropathological changes.

The reviewer raised a comment that “C57BL/6 mice are particularly resistant to the effects of KA”. The reviewer is correct that inbred strains of mice differ significantly in their response to kainate-induced cell death (Neuroscience. 2003;122(2):551-61 // Eur J Neurosci. 2006;24(8):2191-2202 // Exp Neurol. 2002;178(2):219-35). Of note, C57BL/6 mice are relatively resistant to cell death following systemic administration of KA as compared to FVB/N mice, yet both strains show similar sensitivity to seizure activity (Epilepsia. 1995;36(3):301-7). In fact, it is not uncommon that the chemical-induced disease models are species- and strain-dependent. Collagen induced-rheumatoid arthritis is another good example (Nat Protoc. 2007;2(5):1269-75 // Immunopharmacol Immunotoxicol. 2018;40(3):193-200). However, this may not deny the usefulness of C57BL/6 mouse model in exploring the molecular, cellular and pharmacological mechanisms underlying epileptogenesis because KA-induced seizure model can be established with C57BL/6 mice by adjusting KA dose and treatment time. In fact, C57BL/6 mice have been widely used for construction of KA-induced seizure models in numerous studies (e.g., Nat Med. 2012;18(7):1087-94 // Neuron. 2017;95(1):92-105.e5 // Nat Med. 2010;16(4):413-9).

Results p.8 lines 5-7 It is stated that Rev-Erbβ is not likely involved in seizures given the supplemental data shown in Fig.5. However, that figure shows the results of a single seizure paradigm (acute seizures) where only 1 dose of KA was tested. Thus, the statement on p.8 is not completely justified. Similarly, the statements on p.15 lines 13-17 also require modification in light of this criticism.

Response: We accept the reviewer’s criticism that the statements regarding the role of Rev-erbβ in epileptic seizures were too “strong”. Accordingly, the relevant statements have been softened (shown below).

“It was noteworthy that genetic deletion of Rev-erb β (a Rev-erb α paralogue) might have no effects on KA-induced acute seizures (only one KA dose was tested), probably precluding a role of this receptor in regulating seizures (Fig. S7).”

“However, it is argued that the antiepileptic effect of SR8278 might be attributed to antagonism of Rev-erb α because the role of Rev-erb β in epileptic seizures may be limited (Fig. S7)”

Results: What other genes does Rev-Erba regulate besides E4BP4 that could impact seizure susceptibility? What other genes besides the Slc6a1 and Slc6a11 does E4bp4 regulate which could also impact seizure susceptibility?

Response: It seems that the reviewer is concerned about the regulatory pathway of seizure susceptibility by Rev-erb α : Rev-erb α →E4bp4→Slc6a1/Slc6a11→GABAergic function→epileptic seizures. In this study, we first established a tight connection between Rev-erb α and seizure propensity in multiple mouse models of seizures, using both transgenic and small molecule agonist and antagonist approaches. Thereafter, whole cell patch clamp electrophysiology demonstrated changes in GABA-mediated inhibition. By comparing transcriptomes of WT and Rev-erb α KO mice we found dysregulation of Slc6a1 and Slc6a11, two transporters responsible for GABA reuptake. Knockout and knockdown of Rev-erb α confirmed that the GABA concentration is higher in WT neurons and brain slices. By knocking out or overexpressing Rev-erb α , neurons were found to have decreased and increased levels (respectively) of both transporters in a dose-dependent manner. Further, Rev-erb α was found to be responsible for increasing the transporters through its inhibition of E4bp4 transcription, and the specific genomic binding domains E4bp4 targeted were identified for Slc6a1/Slc6a11. Based on multiple techniques and appropriate experimental design, including in vivo and in vitro physiology, RNA and protein profiling, transgenic and small molecule Rev-erb α manipulations, tissue culture, multiple seizure models, and rational

screening, we were able to conclude that circadian changes in Rev-erba expression impact seizures by regulating E4bp4 repression of Slc6a1/Slc6a11, therefore altering GABA reuptake, with associated changes in inhibitory synaptic transmission. This conclusion had been corroborated by the findings that (1) changes in GABAergic function in *Rev-erba*^{-/-} mice were lost when chemical inhibitors (NO 711 and SNAP 5114) of Slc6a1 and Slc6a11 were applied (Fig. S13,14); (2) knockdown of E4bp4 abolished the activation effects of Rev-erba on Slc6a1 and Slc6a11 transcription (Fig. 7H); and (3) E4bp4 ablation sensitized mice to KA-induced acute seizure (i.e., exacerbated seizure severity in *E4bp4*^{-/-} mice as compared to wild-type mice) (Fig. 7I,J). Based on the above analysis, we may provide answers to the questions raised by the reviewer: there is no experimental evidence that other genes besides *E4bp4* and Slc6a1/Slc6a11 are involved in Rev-erba regulation of epileptic seizures.

Results: Were EAATs evaluated? If not, the premise that the role of glutamate in this model can be excluded (e.g. p.11 line 8) is not justified.

Response: We agree with the reviewer that the expression of excitatory amino acid transporters (EAATs) should be evaluated. Accordingly, we have performed new qPCR assays to determine the expression of EAATs (including GLAST, GLT-1, EAAC1, EAAT4 and EAAT5). The results showed that the expression levels of those EAATs were unaffected in Rev-erba deficient mice as compared to wild-type mice. The new data have been added to supplementary Figure 17. Combined with our previous findings that the expression of glutamate receptors and synthetases were unaffected by Rev-erba, we believe that glutamate signaling may not contribute to Rev-erba regulation of epilepsy.

Results/Discussion: It is not clear whether the changes in expression are occurring in glia as well as neurons. If also in glia, a role for alternative mechanisms such as their

role in inflammation or synaptic function must be considered.

Response: We have performed new immunofluorescence experiments to analyze the distribution of Rev-erb α protein. The results show that Rev-erb α is localized in both neurons and glial cells. The new data have been added to Figure 1C-E and supplementary Figure 5 in revised manuscript. We understand the reviewer's concern about whether alternative mechanisms such as glial cell-regulated inflammation and synaptic phagocytosis are involved in Rev-erb α promotion of epileptic seizures. In fact, Rev-erb α has been identified as a repressor of glial activation and neuroinflammation (Proc Natl Acad Sci U S A. 2019;116(11):5102-5107). Loss of Rev-erb α promotes spontaneous neuroinflammation and neuronal dysfunction (Proc Natl Acad Sci U S A. 2019;116(11):5102-5107). Rev-erb α was also shown to restrain synaptic phagocytosis in the brain (bioRxiv, 2020. doi.org/10.1101/2020.05.11.088443). Since glial activation, neuroinflammation and synaptic phagocytosis are contributing factors to the pathogenesis of epilepsy and Rev-erb α is a repressor of these factors, the involvement of inflammation and synaptic phagocytosis in Rev-erb α promotion of epileptic seizures may be rather limited. To alleviate the reviewer's concern, this point has been discussed by adding a sentence (shown below) in the fourth paragraph of DISCUSSION section.

“We may exclude the possibility of additional mechanisms such as inflammation and synaptic phagocytosis contributing to Rev-erb α promotion of epileptic seizures because Rev-erb α is a repressor of glial activation, neuroinflammation, and synaptic phagocytosis.”

Discussion p.16 lines 2-4: This section misses the point that AED transport by p-gp is a controversial topic and to suggest that the Rev-erb ligand regulates AED transport via p-gp requires some data or reference. Thus, this statement should at least be modified to indicate that there are AEDs that are clearly not p-gp substrates and p-gp has not been shown definitively to play a major role in AED resistance.

Response: We have followed the reviewer's suggestion, and modified the statement to indicate that not all AEDs are P-gp substrates (please see the revised manuscript).

Discussion: Prior studies showed that Bmal1 is a transcriptional activator for Rev-erba, and Bmal1 KO mice are seizure-sensitive as well as arrhythmic regarding seizure threshold. More discussion of the interplay between these two molecules is warranted.

Response: We have followed the reviewer's suggestion, and added one paragraph to discuss the role of Bmal1 versus Rev-erba in regulating epileptic seizures (please see the second last paragraph of DISCUSSION, also shown below).

“In a prior report, Bmal1-deficient mice are seizure-sensitive based on measurements of electrical seizure thresholds, although the underlying mechanisms were unexplored. This may suggest a protective effect of Bmal1 on epileptic seizures. The distinct roles of Bmal1 and Rev-erba in regulating seizure susceptibility and other disorders [e.g., hepatitis C virus (HCV) infection] conform to their feedback loop association which is known to be essential for generation of circadian rhythms. On the other hand, the protective role of Bmal1 may suggest involvement of alternative mechanisms independent of the Rev-erba mechanism in regulating epileptic seizures considering Bmal1 is a transcriptional activator of Rev-erba. This is possible because Bmal1 and Rev-erba are showed to regulate biological processes via distinct pathways. For instance, Bmal1 regulates HCV replication through modulating viral receptors CD81 and claudin-1, whereas Rev-erba regulates HCV replication via modulating SCD and subsequent release of infectious particles.”

Responses to reviewer #3

This study examines the expression of the CLOCK component Rev-erbalpha in human tissue resected from patients with temporal lobe epilepsy, and then attempts to establish a connection between this altered expression and seizure propensity in multiple mouse models of seizures, using both transgenic and small molecule agonist and antagonist approaches. The authors conclude that circadian changes in Rev-erbalpha expression may influence seizures by altering GABA uptake, with associated changes in inhibitory synaptic transmission. In general, I find the manuscript to be interesting and, for the most part, competently conducted. I have several suggestions for revision:

The study is very ambitious and uses multiple techniques in a complicated experimental approach, including in vivo and in vitro physiology, RNA and protein profiling, transgenic and small molecule drug Rev-erbalpha manipulations, tissue culture, multiple seizure models, etc. This ambition is both a strength and a weakness of the MS, as the breadth of the approach precludes in depth analyses and follow-up on important findings.

The information concerning Rev-erbalpha expression in human tissue resected from patients with temporal lobe epilepsy is critical to the premise of the whole manuscript, and one of the few direct putative linkers between the CLOCK system and TLE. As presented, there are serious issues with interpretation of this RNA and protein expression data. As is common and depicted in the patient data table (supp. Table 1), it frequently takes decades between initial emergence of seizures and surgical resection, during which time extensive pathology builds in the affected tissues, including extensive neuronal loss, gliosis, inflammation, and other changes. Loss of entire neuronal components of the hippocampus is not uncommon. In one study cited by the authors (Li et al., 2017), CLOCK is expressed only in neurons (and not in GFAP positive astrocytes) in human brain. Several studies have also implicated inflammation and CLOCK gene function as strongly interrelated. Given that the neuronal contribution

to the tissue harvested for RNA and protein profiling undoubtedly had a reduced neuronal component relative to astrocytes, that there was enhanced inflammation in the tissue, and that these changes were restricted to the TLE (and not the brain tumor) tissue, it becomes very hard to interpret the data depicted in Fig. 1A—D without cell specific profiling.

Response: We accept the reviewer's criticism that cell specific profiling of Rev-erba expression is essential for data interpretation. Accordingly, we have performed new immunofluorescence experiments to analyze cell distribution of Rev-erba protein. The results show that Rev-erba is localized in both neurons and glial cells. The new data have been added to Figure 1C-E and supplementary Figure 5 in revised manuscript.

The systemic KA model utilized in much of the manuscript generates data that is also hard to interpret. This is unfortunate since the primary putative linkage between circadian rhythmicity and seizures is derived from studies using this manipulation. Studies like those in the present MS usually profile seizure thresholds to assess pro- (or anti-) seizure effects of experimental manipulations. Injecting a suprathreshold systemic convulsant, and then profiling the seizure stage ends up convolving numerous contributing factors (e.g. drug metabolism, BBB permeability, seizure generalization mechanisms, seizure propensity, mortality, etc.) into the analysis, any of which could contribute to altering seizure stage.

Response: The reviewer is concerned about the systemic KA model used in this study, although this concern may be tempered by use of different seizure testing paradigms involving two mechanistically distinct agents (KA and pilocarpine) as well as a non-pharmacological model involving electrical kindling. The KA model of TLE has greatly contributed to the understanding of the molecular, cellular and pharmacological mechanisms underlying epileptogenesis and ictogenesis. This model presents with neuropathological and electroencephalographic features that are seen in patients with

TLE (Neurosci Biobehav Rev. 2013;37(10 Pt 2):2887-2899). The KA model is regarded as a highly isomorphic model of the human disease, independently of the method of the administration (Neurosci Biobehav Rev. 2013;37(10 Pt 2):2887-2899). The intracerebral and systemic procedures yield similar results in terms of latency to SE, behavioral symptoms, duration of the latent period, and electroencephalographic features of the latent and chronic periods (Neurosci Biobehav Rev. 2013;37(10 Pt 2):2887-2899). However, the systemic KA model has some advantages when compared to the intracerebral administration of KA. For the systemic KA model, many animals can be injected at one time and it does not require from the experimenter to perform surgical procedures, which therefore eliminates post-surgical complications that could affect the animal's health or damage to the brain tissue made by the cannula.

We accept the reviewer's criticism that some confounding factors (such as bioavailability of KA in the brain, and KA receptors) should be considered in data interpretation of Rev-erb α effects with the systemic KA model. Accordingly, we have performed new experiments to determine the pharmacokinetics and brain distribution (hippocampus and cortex) of KA in Rev-erb α KO versus WT control mice after systemic administration. The results showed that the pharmacokinetics and brain distribution of KA (reflective of KA metabolism/biotransformation and BBB permeability) were unaffected in Rev-erb α KO mice. We have also performed new qPCR assays to determine expression of KA receptors [Gluk1, Gluk2, Gluk4 and Gluk5; through which KA induces excitotoxic damage (referred to the seizure generation mechanism)] in the hippocampus and cortex of Rev-erb α KO versus WT mice. We observed no changes in expression levels of KA receptors in the KO mice, suggesting no alterations in the excitotoxic targets of KA. The new data have been added to supplementary Figure 15 in revised manuscript. Therefore, drug metabolism, BBB permeability, and seizure generalization mechanism as potential contributing factors may be excluded.

The authors have a difficulty in understanding how "seizure propensity" and "mortality" confound the data interpretation. "Seizure propensity" and "mortality" seem to be the

terms describing animal phenotypes, but not causal biological factors of KA seizure susceptibility. We did observe attenuated seizures in Rev-erb α KO mice with a lower seizure-associated mortality rate following systemic administration of KA. The data have been added to Figure 2I in revised manuscript.

The most applicable TLE model utilized in this study was the pilocarpine model of spontaneous seizures, depicted in SFig 8. This model generates mice with spontaneous seizures which involve the hippocampus, and a pathological profile which recapitulates important aspects of human TLE (mesial temporal sclerosis). Broadening the assessment of Rev-erb α effects on this model would improve the MS.

Response: We have followed the reviewer's suggestion, and have broadened the assessment of Rev-erb α effects with the pilocarpine model. To be specific, (1) additional measurements (including FJB and TUNEL staining, and seizure frequency) of pilocarpine-induced seizures have been added to previous antagonism (SR8278) experiment (previous SFig 8 and supplementary Figure 10 in revised version); (2) we have tested and compared the susceptibility of Rev-erb α KO and WT mice to pilocarpine-induced seizure (data added to supplementary Figure 11); (3) we have determined the effect of Rev-erb α activation (by SR9009) on pilocarpine-induced seizure in WT mice (data added to supplementary Figure 11). All these new data were consistent with a seizure-promoting effect of Rev-erb α .

However, some aspects of the spontaneous seizure data are curious. Pilocarpine mice have spontaneous seizures in 9-10 day clusters, separated by a week or more during which there are few or no seizures. This seizure biology increases the variability in aggregate seizure plots like those in SFig 8 greatly (since mice do not usually synchronize across populations), and often necessitates depiction of data in a more accessible format, i.e. individual mouse data for the two populations. The small error

bars depicted in this figure are hard to reconcile with this typical manifestation of seizures in this mouse model.

Response: We accept the reviewer's criticism that spontaneous seizure data should be depicted in a more accessible format. Accordingly, the data have been re-plotted to present the daily frequency of spontaneous recurrent seizures in all individual mice (please see new supplementary Figure 10).

The sIPSC effects in Fig. 5 are fairly subtle, and the frequency effects are hard to interpret. If in fact increased GABA uptake is the explanation for the sIPSC changes, then I would expect an enhancement in sIPSC decay time (albeit a more pronounced change than the few % change depicted), but the frequency effects are difficult to interpret. This suggests enhanced interneuron firing, but due to what? Perhaps augmenting this data with recordings of tonic GABA currents would enhance support for the putative linkage between transporter expression and inhibitory function.

Response: The reviewer raised a comment that the change in sIPSC decay time is "subtle". We would like to stress that the changes in both sIPSC decay time and frequency are statistically significant, and the significant p values have been added to the graphs (please see the new Figure 5). The reviewer's judgment seems to be made based on data comparison with a prior study by Li et al (Neuron. 2017;96(2):387-401.e6). Relatively speaking, changes in sIPSC decay time or rise time are "subtle" when compared to changes in sIPSC frequency and amplitude in both studies. According to the Figure 6 of Li study (Neuron. 2017;96(2):387-401.e6) and other studies (Pharmacol Res Perspect. 2017;5(4):e00322 // Front Mol Neurosci. 2019;12:15), the percent of changes in sIPSC decay time and rise time were in the range of 5%-15%. In our study, the percent of change in sIPSC decay time were 12% (Figure 5C) and 14% (Figure 5F). Apparently, the magnitude of changes in sIPSC decay time is consistent between two studies. Therefore, the "subtle" changes in the

time parameters of sIPSC may be normal, and are adequately correlated with alterations in GABAergic function.

We agree with the reviewer that recordings of tonic GABA currents would augment the sIPSC data. Accordingly, we have measured the tonic GABA currents in dentate gyrus granule cells derived from Rev-erb α KO and WT mice. We observed a significant increase in tonic GABA currents in Rev-erb α KO mice as compared to WT mice. As noted by the reviewer, the altered tonic GABA current may account for the increased frequency (i.e., enhanced interneuron firing). The new data have been added to supplementary Figure 13 and the relevant result description to the main text.

As it stands, the subtle effects described on sIPSCs seems somewhat at odds with the much greater seizure effects described in the remainder of the MS. In addition, these effects differ at least in part from the Li et al., 2017 findings when CLOCK was deleted in glutamatergic neurons, where sIPSC effects were much more pronounced and included amplitude, rise and decay time, and frequency changes. This should at least be discussed.

Response: The reviewer raised a comment that the sIPSC effects were much more pronounced in a prior study by Li et al as compared to the effects in current study. We may respectively disagree with the reviewer that the sIPSC effects were much more pronounced in the Li study (Neuron. 2017;96(2):387-401.e6). The sIPSC effects in Li study appear to vary with species, that were much less significant in human tissues (please see the Figure 6 in Li study). In addition, according to the Figure 6 of Li study (Neuron. 2017;96(2):387-401.e6), the percent of changes in sIPSC parameters (decay time, rise time, frequency, and amplitude) were \leq 20%. In our study, the percents of change in sIPSC decay time were about 13% (Figure 5) and the percents of change in frequency were about 40% (Figure 5).

In addition, physiological studies should provide data about cellular properties such as input resistance, membrane potential, firing properties, etc. that are routinely recorded during these types of studies. This could be included as supplemental data, but its absence is an issue.

Response: We have followed the reviewer's suggestion and provided the cellular properties in new supplementary Table 1.

Supplemental Figure 3 is not informative. Perhaps supplemental videos would improve presentation of the finding.

Response: Revised as suggested (please see supplementary videos).

I did not understand the utility of the jet lag data (supplemental figure 4) and how this contributed to the overall development of the MS.

Response: The reviewer raised a concern about the usefulness of the jet lag model. In fact, chronic jet lag is a means of Rev-erb α manipulation, and Rev-erb α is significantly down-regulated in the chronic jet lag model (previous supplementary Figure 4). In order to pinpoint the Rev-erb α effects on epileptic seizures, we also tested and compared the susceptibility of jet-lagged mice and normal mice to seizures. The findings support a seizure-promoting effect of Rev-erb α . The authors understand that the jet lag data are not the main findings, thus they are put into the supplementary data.

Reviewers' Comments:

Reviewer #1:

Remarks to the Author:

This manuscript should be accepted. The authors have answered my concerns and this work should be published. It is a beautiful and substantial piece of work.

I would ask that they provide their ephys data in the supplemental materials as scatter plots the same way they did for the main manuscript.

Reviewer #2:

Remarks to the Author:

The authors provided a vast amount of new data and related information to enhance the manuscript. However, there remain several important weaknesses which should be addressed.

There are as follows:

Please provide detailed genotyping information on REV-ERB-alpha KO mice that allowed distinction of WT, Het and null mice.

Contrary to the authors' response, there is no single universally accepted definition of status epilepticus in mice. They should describe the specific characteristics of the seizure response that defined it in this study.

Statistical comparison of seizure stages requires a non-parametric test since the Racine scale is descriptive and does not represent a continuous ordinal set of values. This impacts the results presented in Figs. 2A, 2G, 3A, 3G, 4B, 4F and 7H

Reviewer #3:

Remarks to the Author:

The authors have addressed my concerns in the revised MS

We wish to thank the reviewers again for careful and valuable reviews. Below are our point-by-point responses.

Responses to reviewer #1

This manuscript should be accepted. The authors have answered my concerns and this work should be published. It is a beautiful and substantial piece of work.

I would ask that they provide their ephys data in the supplemental materials as scatter plots the same way they did for the main manuscript.

Response: Revised as suggested. Thanks.

Responses to Reviewer #2

The authors provided a vast amount of new data and related information to enhance the manuscript. However, there remain several important weaknesses which should be addressed. There are as follows:

Please provide detailed genotyping information on REV-ERB-alpha KO mice that allowed distinction of WT, Het and null mice.

Response: We have followed the reviewer's suggestion, and provided PCR genotyping information on Rev-erb α KO mice (please see new supplementary Figure 20, and the methods section). A 516 bp fragment is indicative of wild-type allele, and a 920 bp fragment is indicative of null allele. Heterozygotes show the fragments of both 516 and 920 bp.

Contrary to the authors' response, there is no single universally accepted definition of

status epilepticus in mice. They should describe the specific characteristics of the seizure response that defined it in this study.

Response: We accept the reviewer's criticism. Accordingly, we have specifically defined status epilepticus (SE) in the revised manuscript (please see the methods section, under "*KA-induced acute seizure model*").

Statistical comparison of seizure stages requires a non-parametric test since the Racine scale is descriptive and does not represent a continuous ordinal set of values. This impacts the results presented in Figs.2A, 2G, 3A, 3G, 4B, 4F and 7H.

Response: We accept the reviewer's criticism. Accordingly, the relevant data have been re-analyzed using Kruskal-Wallis test.

Responses to reviewer #3

The authors have addressed my concerns in the revised MS.

Response: N/A.

Reviewers' Comments:

Reviewer #2:

Remarks to the Author:

My comments were satisfactorily addressed.

Thank you, Thomas N. Ferraro, PhD

Responses to reviewer #2

Comments from reviewer #2

My comments were satisfactorily addressed. Thank you, Thomas N. Ferraro, PhD

Response: N/A